PREPARED FOR SUBMISSION TO JHEP

# Finite Landscape of 6d $\mathcal{N}$ =(1,0) Supergravity

**Hee-Cheol Kim**[a,b] **Cumrun Vafa**[b] **and Kai Xu**[b]

[b]*Department of Physics, POSTECH, Pohang 37673, Korea*

[c]*Jefferson Physical Laboratory, Harvard University, Cambridge, MA 02138, USA*

ABSTRACT: We present a bottom-up argument showing that the number of massless fields in six-dimensional quantum gravitational theories with eight supercharges is uniformly bounded. Specifically, we show that the number of tensor multiplets is bounded by $T \leq 193$, and the rank of the gauge group is restricted to $r(V) \leq 480$. Given that F-theory compactifications on elliptic CY 3-folds are a subset, this provides a bound on the Hodge numbers of elliptic CY 3-folds: $h^{1,1}(\mathrm{CY}_3) \leq 491$, $h^{1,1}(\mathrm{Base}) \leq 194$ which are saturated by special elliptic CY 3-folds. This establishes that our bounds are sharp and also provides further evidence for the string lamppost principle. These results are derived by a comprehensive examination of the boundaries of the tensor moduli branch, showing that any consistent supergravity theory with $T \neq 0$ must include a BPS string in its spectrum corresponding to a "little string theory" (LST) or a critical heterotic string. From this tensor branch analysis, we establish a containment relationship between SCFTs and LSTs embedded within a gravitational theory. Combined with the classification of 6d SCFTs and LSTs, this then leads to the above bounds. Together with previous works, this establishes the finiteness of the supergravity landscape for $d \geq 6$.

## 1   Introduction

After many decades of investigations in string theory, the set of known consistent solutions to string theory (with a given cutoff) is finite. The finiteness of string vacua based on Calabi-Yau manifolds was conjectured long ago by Yau [1] and the idea that the number of

quantum gravity vacua should be finite was suggested independently in [2, 3] and further refinements of this idea have been discussed in [4, 5]. Indeed this finiteness is at the heart of the motivation for the Swampland program (see e.g. [6] for a recent review), as it suggests the existence of unexpected consistency requirements that trim the a priori infinite set of apparently consistent possibilities to a finite set.

Since this finiteness is such an important cornerstone of the Swampland principle, it is imperative to seek string-independent arguments which support it. If the finiteness can be proven using only general principles of consistency of quantum gravity theories and a few Swampland principles, we can gain confidence about the relative completeness of our understanding of these principles. Moreover, it would be important to check whether the resulting finiteness bounds are saturated with the known examples from string theory. A natural approach to testing finiteness would be to begin with the most supersymmetric cases and, at the very least, establish bounds on the number of allowed massless fields in the theory. For theories with 32 supercharges, supersymmetry is strong enough to completely fix all possible configurations. For the next class, with 16 supercharges, it has been shown in [7] that in $d = 10$ and in the non-chiral cases, the possibilities are finite because the rank of the gauge group must satisfy $r(G) \leq 26 - d$, and for the chiral case in 6d with $\mathcal{N} = (2, 0)$ supersymmetry, anomaly fixes the massless matter content.

The next class to study would be the case with 8 supercharges, which is first encountered in 6d theories with $\mathcal{N} = (1, 0)$ supersymmetry. Since this theory is chiral, anomaly cancellation imposes constraints on the structure of the allowed theories. However, these constraints are too weak to restrict the set of allowed massless fields to a finite number. Nevertheless, various consistency requirements in quantum gravity provide growing evidence for the finiteness of such theories (see for example [8–11]), although no complete argument has yet been formulated to show the finiteness of allowed massless fields. In this paper, we aim to fill this gap by arguing that there is a universal bound on the number of massless fields for each type of matter, namely tensors $T$, vectors $V$, and hypers $H$.

The basic strategy is as follows: we focus on tensor branch, where the moduli space is controlled by scalar vacuum expectation values (vevs) in the tensor multiplets. The local geometry of these scalars is fixed by supersymmetry. To fully characterize the theory, we must identify the boundaries of the scalar vevs. From the viewpoint of effective field theory (EFT), the only plausible reason for the existence of boundaries in the tensor moduli space is the emergence of massless modes that were not accounted for, occurring at certain points in the moduli space. In other words, boundaries arise when a new local conformal or little string sector emerges. We will assume that such boundaries only involve the appearance of superconformal theories, or in the case of infinite distance, fundamental or little string theories (as suggested by the emergent string conjecture [12]). These theories at boundaries have been classified [13–17], and notably, they require the presence of tensionless BPS strings at these loci. The moduli space of the tensor branch is defined by the scalar vevs under which the tension of all these strings are positive. We employ a series of arguments based on the existence of supergravity black string solutions for BPS strings with specific charges to suggest that among the BPS strings, there must exist at least one distinguished class, which we call an '*H-string*', corresponding either to the perturbative heterotic string

or to a little string made of the strings that can, at least individually, become tensionless. Using this, together with the classification of little strings and the unitarity constraints on the $H$-string, we derive the bounds on $T$, $H$ and $V$. When comparing these bounds with F-theory vacua on elliptic Calabi-Yau 3-folds, we find that specific Calabi-Yau manifolds saturate these bounds. This reinforces the belief in the string lamppost principle (SLP) (or what is sometimes referred to as 'string universality' [18]), which suggests that all consistent quantum gravity backgrounds have a string theory realization.

The organization of this paper is as follows. Section 2 provides a review of the basics of 6d $\mathcal{N} = (1,0)$ supergravity theories, focusing on the consistency conditions imposed by anomaly constraints and the unitarity of BPS string states. In Section 3, we prove the finiteness of massless fields in 6d supergravity by analyzing the boundaries of tensor moduli space and exploring properties of BPS strings, particularly $H$-string, with vanishing tension. In Section 4, precise bounds on massless fields are derived, and some Swampland examples with finite and small numbers of fields are discussed. Section 5 offers geometric interpretations for the structures of 6d supergravity theories uncovered in this study. Finally, Section 6 concludes with a brief summary and further discussions. Some technical discussions are presented in the Appendices.

## 2 Overview of 6d $\mathcal{N} = (1,0)$ Supergravity Theories

In this section, we review the key features of 6d $\mathcal{N} = (1,0)$ supergravity theories and outline the known consistency conditions for the field theory sector of the low-energy theory. This includes a brief discussion on tensor branch and the anomaly cancellation mechanism. Additionally, we introduce a special class of BPS strings closely associated with heterotic strings, which will play key roles in our discussions.

### 2.1 Supergravity and tensor moduli space

In 6d $\mathcal{N} = (1,0)$ supergravity, the massless field content consists of a gravity multiplet along with three types of matter multiplets: $T$ tensor multiplets, $V$ vector multiplets for gauge group $G$ (where $V = \dim G$), and $H$ hypermultiplets in representations of $G$ [19–22]. The effective field theory with this matter content is heavily constrained by supersymmetry and anomaly cancellation conditions.

The gravity multiplet includes a self-dual 2-form gauge field $B_{\mu\nu}^+$, while each tensor multiplet contains an anti-self-dual 2-form field $B_{\mu\nu}^-$ and a real scalar field. The charged objects corresponding to these 2-form fields are 2-dimensional strings, whose charges reside in a charge lattice $\Gamma \subset \mathbb{R}^{1,T}$. The intersection form $\Omega_{\alpha\beta}$ for these 2-form charges is a metric with signature $(1_+, T_-)$, and it defines an integral inner product between any two charge vectors $v, w \in \Gamma$ as $v \cdot w = \Omega_{\alpha\beta} v^\alpha w^\beta \in \mathbf{Z}$. In 6d supergravity, the string charge lattice must be a unimodular lattice, meaning that $\Gamma = \Gamma^*$. This result can be derived using a compactification on $\mathbb{CP}^2$ to 2d, as introduced in [23]. Under this reduction, each (anti-)self-dual 2-form field becomes a chiral scalar field in 2d. The modular invariance of the torus partition function in the resulting 2d theory requires the charge lattice to be unimodular, which, in turn, imposes that the 6d charge lattice $\Gamma$ is unimodular.

The real scalar fields in the tensor multiplets are collectively represented by a vector $J \in \mathbb{R}^{1,T}$ subject to the constraint $J^0 > 0$ and $J \cdot J = 1$. These parametrize the moduli space of the tensor branch, whose geometry reflected by their kinetic term is a coset space $SO(1,T)/SO(T)$ with a locally hyperbolic metric (except for $T = 1$ case where it is flat). As we will see, not all $J \in SO(1,T)/SO(T)$ are physically allowed, only a subset $\hat{\mathcal{M}} \subseteq SO(1,T)/SO(T)$ corresponds to physical vacua; moreover, different $J \in \mathbb{R}^{1,T}$ may give rise to physically equivalent vacua related by dualities, and we may identify them and get $\mathcal{M} = \hat{\mathcal{M}}/\{\text{dualities}\}$. $\mathcal{M}$ is referred to as the *tensor moduli space* and $\hat{\mathcal{M}}$ is referred to as the *marked tensor moduli space* [24]. In the discussions below, we first focus on 6d $\mathcal{N} = (1,0)$ supergravity theories with non-zero tensor multiplets, so $T \geq 1$, and return to the $T = 0$ case when we put a bound on the number of massless fields. Moreover, when there are dualities in the tensor moduli space, we need to consider the marked version.

The tensor moduli space is bounded by singularities where the effective field theory description breaks down. Such singularities can occur either at finite or infinite distances in the moduli space, and they arise only when certain new light degrees of freedom appear near the singular locus. In 6d (1,0) supersymmetric theory, natural light degrees of freedom are tensionless BPS strings, which correspond to elementary excitations of 6d local SCFTs when the singularity is at a finite distance, or to those of LSTs and critical (heterotic or Type II) strings when the singularity is at an infinite distance. From now on, we will assume that

> *Boundaries of tensor moduli are marked by emergence of tensionless BPS strings. Finite distance boundaries correspond to SCFT points and infinite distance ones correspond to LST's or critical strings.*

This is one of the two main assumptions in our proof of the finiteness of 6d supergravity. The second assumption, regarding the completeness of the 6d SCFT/LST classification, will be introduced shortly. The tensor moduli space is therefore defined as a subspace of the coset space with boundaries, expressed as

$$J \cdot Q_i \geq 0 \quad \text{with} \quad J \cdot J = 1 \, , \tag{2.1}$$

where $Q_i$ represents the charges of all BPS strings. It is important to note that the tension $T_Q$ of a BPS string with charge $Q$ is determined by the moduli vector via $T_Q \sim J \cdot Q$. This inequality ensures the unitarity condition for BPS strings that requires the tensions of all BPS strings to be non-negative within the tensor moduli space.

Thus, the structure of the tensor moduli space is dictated by the spectrum of BPS strings, whose charges populate the so-called *BPS cone* within the charge lattice $\Gamma$. We define the BPS cone as a cone of BPS string states generated by *primitive BPS strings* within a tensor (or string) charge lattice $\Gamma$. Primitive BPS strings are those whose tensor charge cannot be written as a non-negative linear combination of the tensor charges of other BPS strings. The reason for having a cone structure is that positive linear combinations of BPS strings preserve the same supersymmetry. The tensor charges $\mathcal{C}_i$ of primitive BPS

strings, or the strings themselves, will be called the *generators of the BPS cone.*[1] The BPS cone is, therefore, the set of all non-negative integral combinations of these generators. Any string, or its tensor charge, inside the BPS cone in the lattice $\Gamma$, is referred to as *effective*.

The dual cone to the BPS cone is called the *tensor cone* denoted by $\mathcal{T}$. Specifically, the tensor cone is the space of the moduli vector $J$ satisfying

$$J \cdot \mathcal{C}_i \geq 0 \quad \text{with} \quad J^2 \geq 0 , \tag{2.2}$$

for all generators $\mathcal{C}_i$. The (marked) tensor moduli space is a hyperbolic slice of this tensor cone defined by the constraint $J^2 = 1$. This is equivalent to the definition (2.1) as the $Q_i$ are generated by the positive combinations of $\mathcal{C}_i$.

We can also prove that the moduli vector $J$ lies in the BPS cone, and thus can be written as

$$J = \sum_i a_i \mathcal{C}_i , \tag{2.3}$$

with $a_i \geq 0$ as coefficients. The inverse is not necessarily true: not all positive combinations of $\mathcal{C}_i$ lead to an allowed $J$ of the tensor branch. To prove this, we start by choosing a moduli vector $J$ at any arbitrary point in the tensor cone. Because $J$ is a time-like vector with $J^2 \geq 0$ and $J^0 > 0$, it must intersect positively with any other moduli vector, say $J'$. As a result, $J' \cdot J \geq 0$ holds for all $J'$ throughout the tensor moduli space, thereby confirming that $J$ is in the BPS cone and satisfies the relation above. An additional proof using BPS black strings will be presented in Section 2.4.

We now address a relation between the BPS cone generators and the intersections of tensor multiplets in 6d supergravity. The definition of the tensor cone $\mathcal{T}$ in (2.2) tells us that every generator of the BPS cone with $T \geq 1$ is related to a "*shrinkable*" tensor multiplet, whose BPS strings become tensionless at the boundary where $J \cdot \mathcal{C}_i = 0$. In six-dimensional supergravity, the only BPS strings that can become tensionless inside the tensor moduli space are SCFT strings, little strings or critical strings. A notable feature of these shrinkable strings is that their tensor charges have a non-positive norm. We thus find that all generators in the tensor moduli space must obey

$$\mathcal{C}_i^2 \leq 0 , \tag{2.4}$$

and correspond to SCFT strings[2], little strings or critical strings. Hence, we can determine the properties of the generators and their relationships, such as the types of generators and their intersections, by utilizing the classification of SCFTs and LSTs that we summarize in Section 3.4 and Appendix A. An immediate conclusion we can deduce from the classification of SCFTs and LSTs is that the minimal charge for each shrinkable generator $\mathcal{C}_i$ is always realized by a BPS string. Therefore, when we refer to a generator $\mathcal{C}_i$ with $\mathcal{C}_i^2 \leq 0$, it often implies its BPS string counterpart.

---

[1]It can be proved for F-theory examples that there are finitely many generators up to the action of duality group [25].

[2]In particular, every instantonic BPS string in SCFTs that carries a unit instanton number is always primitive and thus becomes a generator.

| $G_i$ | $SU(N)$ | $SO(N)$ | $Sp(N)$ | $G_2$ | $F_4$ | $E_6$ | $E_7$ | $E_8$ |
|---|---|---|---|---|---|---|---|---|
| $\lambda_i$ | 1 | 2 | 1 | 2 | 6 | 6 | 12 | 60 |

**Table 1**. Normalization factors for gauge groups

## 2.2 Anomaly cancellation

Local anomalies, such as gravitational and gauge anomalies coming from massless chiral fields, can be canceled via the Green-Schwarz-Sagnotti mechanism [22, 26] provided the 1-loop anomaly polynomial $I_8$ factorizes as

$$I_8 = \frac{1}{2}\Omega_{\alpha\beta}\, X_4^\alpha\, X_4^\beta \;, \quad X_4^\alpha = -\frac{1}{2}b_0^\alpha \,\mathrm{tr}R^2 + \frac{1}{4}\sum_i b_i^\alpha \frac{2}{\lambda_i}\mathrm{tr}F_i^2 \;, \tag{2.5}$$

with anomaly vectors $b_0, b_i \in \mathbb{R}^{1,T}$, the curvature 2-form $R$, and the gauge field strength $F_i$ for a gauge group $G_i$, where $i$ runs over each gauge group factors. The normalization factors $\lambda_i$ are summarized in Table 1. It then follows that the addition of the Green-Schwarz (GS) term to the effective Lagrangian results in a modification of the Bianchi identities for the 2-form fields as follows:

$$\Delta\mathcal{L} \sim \Omega_{\alpha\beta}B^\alpha \wedge X_4^\beta \quad \rightarrow \quad dH^\alpha = X_4^\alpha \;, \tag{2.6}$$

where $H^\alpha$ represents the field strength for the 2-form field $B^\alpha$. The factorization of $I_8$ can occur, and thus anomalies can be canceled through GS-terms, if the following conditions are satisfied:

$$H - V = 273 - 29T \;, \quad b_0 \cdot b_0 = 9 - T \;,$$

$$B_{\mathbf{adj}}^i = \sum_{\mathbf{r}} n_{\mathbf{r}}^i B_{\mathbf{r}}^i \;, \quad b_0 \cdot b_i = \frac{\lambda_i}{6}\left(\sum_{\mathbf{r}} n_{\mathbf{r}}^i A_{\mathbf{r}}^i - A_{\mathbf{adj}}^i\right) \;,$$

$$b_i \cdot b_i = \frac{\lambda_i^2}{3}\left(\sum_{\mathbf{r}} n_{\mathbf{r}}^i C_{\mathbf{r}}^i - C_{\mathbf{adj}}^i\right) \;, \quad b_i \cdot b_j = 2\lambda_i\lambda_j \sum_{\mathbf{r},\mathbf{s}} n_{\mathbf{r},\mathbf{s}}^{ij} A_{\mathbf{r}}^i A_{\mathbf{s}}^j \quad i \neq j \;, \tag{2.7}$$

where $n_{\mathbf{r}}^i$ denotes the number of matter fields in the representation $\mathbf{r}$ of the gauge group $G_i$, and **adj** denotes refers to the adjoint representation. The trace indices $A_{\mathbf{r}}, B_{\mathbf{r}}, C_{\mathbf{r}}$ are defined as

$$\mathrm{tr}_{\mathbf{r}}F^2 = A_{\mathbf{r}}\mathrm{tr}F^2 \;, \quad \mathrm{tr}_{\mathbf{r}}F^4 = B_{\mathbf{r}}\mathrm{tr}F^4 + C_{\mathbf{r}}(\mathrm{tr}F^2)^2 \;, \tag{2.8}$$

where 'tr' without subscript represents a trace over the fundamental (or defining) representation.

The effective field theories that meet the anomaly cancellation conditions in (2.7) will be referred to as anomaly-free theories. A large class of anomaly-free 6d theories has been constructed in several works, such as [22, 27–32]. Many of these theories are realized in string theory, particularly through F-theory compactifications. See [33] for a detailed review of string theory realizations of 6d supergravity. Furthermore, systematic classifications of

anomaly-free theories with non-Abelian gauge groups have been performed using various approaches, for instance, in [10, 11, 31, 32]. These have also led to the construction of infinite families of anomaly-free theories with arbitrarily large numbers of tensor multiplets. See also [34] for some more examples of such infinite families.

To determine whether an anomaly-free effective field theory is consistent with quantum gravity and ultimately part of the supergravity landscape, new criteria beyond anomaly cancellation seem to be needed. This is a highly challenging task. However, notable progress has been made in this area by utilizing the unitarity of string probes in 6d supergravity. For more details on this and related discussions, see [8–11, 35]. Some of these ideas play a key role for us, as we will note later in the paper.

## 2.3  BPS strings

BPS strings play a crucial role in understanding the quantum aspects of 6d supergravity theories. These are two-dimensional objects charged under 2-form tensor fields $B^{\pm}_{\mu\nu}$, preserving 4 supercharges. At low energies, the worldsheet theory of these strings reduces to a 2d $\mathcal{N} = (0, 4)$ SCFT. These worldsheet SCFTs are characterized by their central charges and the types of current algebras coupled to the 6d bulk gauge symmetry. A particular class of BPS strings, known as *supergravity strings*, plays a significant role in 6d supergravity. The worldsheet $(0, 4)$ SCFT for these strings features a superconformal $SU(2)_R$ R-symmetry that is identified with an $SU(2)$ subgroup of the transverse $SO(4)$ rotation group. The central charges and current algebra levels for these strings can be computed via the anomaly inflow mechanism, as detailed in [8]. For a worldsheet CFT with string charge $Q$, the left- and right-moving central charges $c_L$ and $c_R$ are given by

$$c_L = 3Q \cdot Q + 9b_0 \cdot Q + 2 , \quad c_R = 3Q \cdot Q + 3b_0 \cdot Q . \tag{2.9}$$

Here, we have already subtracted the center of mass contributions, which are $c_L^{\mathrm{com}} = 4, c_R^{\mathrm{com}} = 6$.

On the other hand, there exists a class of strings with $Q^2 < 0$ that arise as BPS excitations in 6d SCFTs or LSTs embedded in supergravity. These strings are shrinkable, with an enhanced $SU(2)_I$ R-symmetry at low energy where their tension vanishes in the tensor moduli space. This enhanced $SU(2)_I$, rather than $SU(2)_R$, serves as the conformal R-symmetry of the $(0, 4)$ superconformal algebra for the worldsheet CFTs. Thus, their central charges differ from those given in (2.9). See [36, 37] for details. However, in the following discussions, we will primarily focus on the supergravity strings with central charges as in (2.9).

The levels $k_L$ and $k_i$ for the $SU(2)_L$ subgroup of the transverse $SO(4)$ rotation and the bulk gauge symmetry $G_i$ are

$$k_L = \frac{1}{2}(Q \cdot Q - b_0 \cdot Q + 2) , \quad k_i = Q \cdot b_i . \tag{2.10}$$

Note that the integrality of $k_L$ implies, in particular, that $b_0 \cdot Q$ is an integer for all BPS charges $Q$. Moreover, since BPS strings span the charge lattice (as follows from the fact that they span at least the tensor cone), it follows that $b_0$ belongs to the dual of the string

lattice charge. But since the string lattice is self-dual, this implies that $b_0$ itself belongs to the lattice.

For a unitary supergravity string, central charges $c_L$, $c_R$ and levels $k_L$, $k_i$ must be non-negative. Specifically, the conditions $c_R \geq 0$ and $k_L \geq 0$ require that

$$Q \cdot Q > 0 \ , \tag{2.11}$$

with three exceptions:

$$\text{1) } Q^2 = -1 \ , \ k_L = 0 \ , \qquad \text{2) } Q^2 = 0 \ , \ k_L = 0 \ , \qquad \text{3) } Q^2 = 0 \ , \ k_L = 1 \ . \tag{2.12}$$

These exceptions are the charge classes of an E-string for case 1), the heterotic string for case 2), and the Type II string for case 3). Interestingly, an E-string appears as a BPS excitation in an SCFT with a '$-1$' tensor multiplet but also acts as a supergravity string. This is possible because, although the SCFT from a '$-1$' tensor multiplet has an enhanced $SU(2)_I$ R-symmetry, the worldsheet 2d SCFT on a single E-string has no degrees of freedom that carry the $SU(2)_I$ charge. Its central charges $c_L = 8$ and $c_R = 0$ agree with the central charge formula for a supergravity string in (2.9).

The current algebras for the symmetry groups $SU(2)_L$ and $G_i$ in the context of the $(0,4)$ superconformal algebra for supergravity strings are realized in the left-moving sector. The realization of the level-$k$ Kac-Moody algebra of group requires a central charge contribution as (see, e.g., [38])

$$c_G = \frac{k \cdot \dim G}{k + h^\vee} \ , \tag{2.13}$$

where $\dim G$ and $h^\vee$ represent the dimension and dual Coxeter number of the group $G$, respectively. For a unitary 2d (0,4) SCFT, this then imposes the following constraint on the 6d gauge groups.

$$\sum_i c_{G_i} + c_{SU(2)_L} \leq c_L \ . \tag{2.14}$$

This constraint has been applied in various works, including [8–11, 35], to show that certain anomaly-free theories involve non-unitary BPS strings violating this constraint, and thus belong to the Swampland.

A distinguished class of BPS strings for the upcoming discussions are those with string charge $f$ satisfying $f^2 = 0$ and $b_0 \cdot f = 2$, corresponding to the second case in (2.12). These strings have central charges

$$c_L = 20 \ , \quad c_R = 6 \ , \quad k_L = 0 \ . \tag{2.15}$$

These are identical to the central charges of 2d CFTs on critical heterotic strings compactified to six dimensions and little strings, which are the fundamental excitations in LSTs. We will prove a crucial fact for our purposes, that *any consistent 6d supergravity theory with tensor multiplets must contain a string of this type (not necessarily uniquely) in its BPS spectrum.* Furthermore, we will show that there always exists an infinite distance limit in

the tensor branch where these strings become tensionless. This limit corresponds exactly to the asymptotic limit with tensionless strings in the Emergent String Conjecture proposed in [39, 40]. Indeed, in this asymptotic limit, a string in the charge class of $f$ will become the critical heterotic string in some duality frame. With this reasoning, we will refer to these particular BPS strings satisfying (2.15) as "*H-strings*". More generally, there may also be certain little strings that share the same charge as the $f$-string[3], which makes the dual heterotic string theory non-perturbative. A novel relationship between these $H$-strings and BPS strings in local SCFTs embedded in supergravity, which we establish in a later section, is key to proving the finiteness of 6d supergravity.

## 2.4 BPS strings and supersymmetric black strings

Supersymmetric black strings in 6d supergravity and their reduction to 5d BPS black holes provide another key feature of the tensor moduli space. BPS black string solutions in 6d (1,0) supergravity are studied through the attractor mechanism in [41]; the attractor mechanism shows that the value of the moduli vector $J$ near the horizon is expressed in terms of the central charges of the black object [42, 43]. For a given charge $Q$, attractor mechanism involves minimizing $J \cdot Q$ subject to $J^2 = 1$. This minimization leads to $J \propto Q$ [41]. In general, $J$ may not be properly quantized. In such cases, for large enough $J$ we can apply a small shift $J \to J + \epsilon$ with $\epsilon \ll J$ to achieve the correct quantization for the black string charge $Q^\alpha$ in the charge lattice $\Gamma$. This allows us to find a BPS black string solution at any point within the tensor moduli space:

$$Q_{B.S}^\alpha = kJ^\alpha \ , \tag{2.16}$$

where $k$ is a large positive constant, determined by $k = \sqrt{\Omega_{\alpha\beta}Q^\alpha Q^\beta}$ with $J^2 = 1$. The charge $Q_{B.S}^\alpha$ of the black string must lies inside the BPS cone. This also means that the moduli vector $J$ must lie inside the BPS cone, as shown in Figure 1. This leads to the relation in (2.3).

It is also important to note that instantons for a gauge group $G_i$ are sources for a 2-form tensor gauge field with charge $b_i$. Moreover, the gauge instanton is a supersymmetric configuration and so $b_i$ is an effective charge. We also assert that $b_0$ resides in the BPS cone. To support this, consider a compactification of the 6d theory to 2d on $K3$ (or half-$K3$)[4]. We claim that this can be chosen to be a supersymmetric compactification. This follows from the strong form of the cobordism conjecture [44], stating that a supersymmetry-preserving defect always exists whenever a cobordism class is killed in a supersymmetric setting. For such a supersymmetric compactification, the gravitational term in the Bianchi identity (2.6) leads to a tadpole contribution of $-24b_0$ (or $-12b_0$); therefore, the assumption of

---

[3]When these little strings support gauge algebras, the 2d SCFTs on them would have two branches: the instantonic branch, which describes the zero modes of gauge instantons, and the heterotic string branch, which corresponds to the zero modes of the dual heterotic string. These two branches are smoothly connected in the worldsheet CFTs.

[4]Even though half-$K3$ is non-compact the asymptotic geometry is a constant torus over $R$ so there would be no contribution from boundary to the gravitational anomaly, which is why we can also use it to get a stronger bound on the multiple of $b_0$.

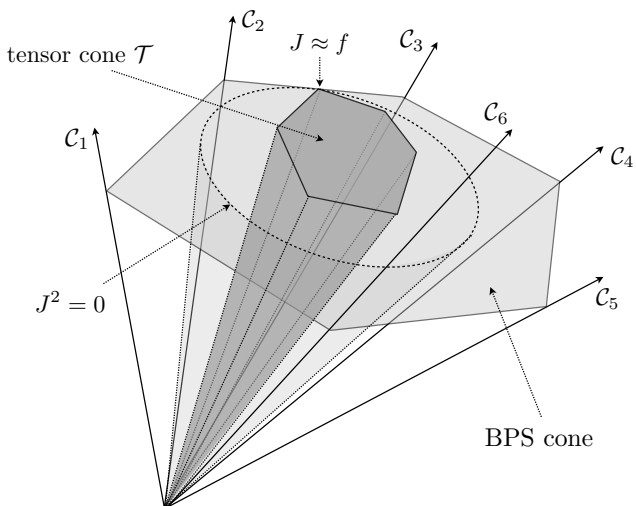

**Figure 1**. The tensor cone $\mathcal{T}$ inside a BPS cone generated by $\mathcal{C}_{1,2,3,4,5,6}$. The dotted circle represents a round cone with $J^2 \geq 0$. The boundary of the tensor cone can meet the boundary of BPS cone only at points where $J \approx f$ with $f^2 = 0$.

existence of a supersymmetric defect requires the existence of a supersymmetric BPS string with a charge $Q = 24b_0$ (or $Q = 12b_0$). We will show that the factor 12 here is optimal in Appendix D, i.e. any smaller multiple of $b_0$ is not necessarily a BPS string. Hence, based on the stronger form of the cobordism conjecture, we assume that a positive integer multiple of $b_0$ is effective. This leads to the statement that

$$J \cdot b_0 \geq 0 \ , \quad J \cdot b_i \geq 0 \tag{2.17}$$

must hold in the tensor moduli space. There are alternative arguments supporting the first positivity: The coefficient of the Gauss-Bonnet term in the effective field theory (obtained via supersymmetry completion of the Green-Schwarz term) is $J \cdot b_0$, and it has been generally argued, based on the unitarity of spectral representation [45] and the positivity of higher-derivative corrections to black hole entropy [46], that the coefficient of GB term is non-negative.

## 3 Proof of Boundedness

In this section, we provide a proof for the boundedness of the number of tensor multiplets and rank of gauge algebras in the context of 6d effective field theories. Our approach relies on the analysis of the tensor moduli space, the detailed structure of the BPS string charge lattice, the unitarity of BPS strings, and the classification of 6d SCFTs and LSTs. For this proof, we now assume that

> *The classification of 6d SCFTs and LSTs is complete.*

This is our second key assumption, in addition to the assumption regarding the boundary of the tensor moduli space discussed in Section 2.1, for the proof. We find this assumption reasonable, as the classifications provided in [13–15, 17] account for all known constructions of 6d SCFTs and LSTs, including the F-theory models, brane systems in string theory, and gauge theoretic constructions, and cover all theories arising from these frameworks. The classification data of SCFTs and LSTs will be extensively used in several key steps of our proof below.

Let us briefly outline our main steps for proving the boundedness of the 6d supergravity landscape. The key steps are as follows:

---

**Claims**

1. Every 6d (1,0) supergravity with $T \geq 1$ (except for a special isolated case with $T = 9$ and $b_0 = 0$) must contain an *H-string* with charge $f$,

$$f^2 = 0 \ , \quad b_0 \cdot f = 2 \ , \quad k_L(f) = 0 \ , \quad f \cdot \mathcal{C}_i \geq 0 \tag{3.1}$$

   for all generators $\mathcal{C}_i$ of the BPS cone.

2. Any generator $\mathcal{C}_i$ with $f \cdot \mathcal{C}_i = 0$ and gauge algebra $\mathfrak{g}_i \notin \{\varnothing, \mathfrak{su}_{2,3,4}, \mathfrak{sp}_2, \mathfrak{g}_2\}$ is an element of a little string theory (LST) for the charge class $f$.

3. The classification of SCFTs and LSTs, along with the unitarity of the *H*-string, places upper bounds on the number of tensor, vector and hypermultiplets.

---

These claims are essential steps necessary for the proof. In the upcoming subsections, we will rigorously prove each claim and subsequently derive the exact upper bounds for the number of tensors and the rank of gauge algebras in six-dimensional effective field theory.

To prove the first claim, we will apply a procedure, which we term "*blowdowns*", which shrink all generators with self-intersection $-1$, and demonstrate that a subspace of the BPS cone after this procedure must contain a string with $f^2 = 0$ and $b_0 \cdot f = 2$ corresponding to an *H-string*, and that this string must have been within the original BPS cone before the blowdowns.[5] A key ingredient for this step is to use the fact that a positive integer multiple of $b_0$ is effective, which we argued earlier.

For the second claim, we first show that the $f$-string we identified has a tensionless limit, and in this limit, the tensions of generators $\mathcal{C}_i$ with $f \cdot \mathcal{C}_i = 0$ and $\mathfrak{g}_i \neq \mathfrak{g}_{\text{small}}$ also vanishe at the same rate as the tension of the $f$-string, where $\mathfrak{g}_{\text{small}} \in \{\varnothing, \mathfrak{su}_{2,3,4}, \mathfrak{sp}_2, \mathfrak{g}_2\}$. Based on this fact, we will then show that these generators reside within the BPS cone of an LST for the $f$-string.

---

[5]It is important to note that, even though the terminology of *blowdowns* is motivated from algebraic geometry realization of these vacua, we will not be making any assumptions about how the 6d supergravity arises, and in particular we will not be assuming a geometric realization of the theory. We use this terminology to avoid proliferation of introducing new terms.

Finally, the last claim will be proven by showing that all but few tensor multiplets supporting non-Higgsable exceptional gauge algebras, which could contribute negatively to the gravitational anomaly, must be part of an LST for the $f$-string and are always accompanied by additional tensor multiplets, known as *interior links* in their classification. The positive contributions from these additional tensor multiplets to the gravitational anomaly ensure that an arbitrarily large number of them is not possible.

Now, we will proceed to detail the steps in the proof of boundedness. First we prove that distinct generators of the BPS cone have a positive inner product with one another. Using this result, we define a 'blowdown' procedure to establish the existence of an $H$-string (i.e., a null BPS string $f$ with $f \cdot b_0 = 2$). Next, we show a containment relation between generators $\mathcal{C}_i$ and LSTs in a charge class $f$ when $f \cdot \mathcal{C}_i = 0$. Finally, we use these results to conclude that the number of tensors is bounded.

## 3.1 $\quad \mathcal{C}_i \cdot \mathcal{C}_j \geq 0$

As the first step, we will demonstrate that all generators of the BPS cone must have non-negative intersections with one another. Specifically, two distinct generators, $\mathcal{C}_i$ and $\mathcal{C}_j$, must satisfy the condition

$$\mathcal{C}_i \cdot \mathcal{C}_j \geq 0 \quad \text{for all } i \neq j . \tag{3.2}$$

We give two completely different proofs, one based mainly on geometry of BPS cone and the other also based on worldsheet theory of BPS strings. When both generators can shrink simultaneously, i.e., when the conditions $J \cdot \mathcal{C}_i = 0$ and $J \cdot \mathcal{C}_j = 0$ can both hold simultaneously in the tensor moduli space, this relation directly follows from the classification of 6d CFTs and LSTs. Therefore, it remains to prove this for cases where the conditions $J \cdot \mathcal{C}_i = 0$ & $J \cdot \mathcal{C}_j = 0$ cannot hold at the same time.

Let us first consider an hyperplane $H$, defined by $Q \cdot \mathcal{C}_j = 0$ for a generator with $\mathcal{C}_j^2 < 0$ in the pair. Here, we have $\mathcal{C}_i^2 \leq 0$ for the other generator, and we will return to the case $\mathcal{C}_i^2 = \mathcal{C}_j^2 = 0$ at the end of this subsection. This hyperplane divides the BPS cone into two regions: one where $Q \cdot \mathcal{C}_j > 0$ and the other where $Q \cdot \mathcal{C}_j < 0$. Now, assume for contradiction that $\mathcal{C}_i \cdot \mathcal{C}_j < 0$. This places $\mathcal{C}_i$ and $\mathcal{C}_j$ on the same side, where $Q \cdot \mathcal{C}_j < 0$, while the tensor cone satisfying $J \cdot \mathcal{C}_j > 0$ lies on the other side. From the fact that $\mathcal{C}_i$ and $\mathcal{C}_j$ are on the same side, and that the BPS cone is convex and locally polyhedral near $\mathcal{C}_j$[6], it then follows that $\mathcal{C}_j$ can always be connected to $\mathcal{C}_i$ through a sequence of generators lying on the same side of the hyperplane arranged in such a way that each consecutive pair of generators shares the same face of the BPS cone, taking the form like $\mathcal{C}_j \to \mathcal{C}_k \to \mathcal{C}_l \to \cdots \to \mathcal{C}_i$, where $\mathcal{C}_j$ and $\mathcal{C}_k$ share a face, $\mathcal{C}_k$ and $\mathcal{C}_l$ share another face, and so forth.[7] One then sees that $\mathcal{C}_k$ lies on the side where $Q \cdot \mathcal{C}_j < 0$, but it also shares the same face of the BPS cone with $\mathcal{C}_j$.

---

[6]As we assumed $\mathcal{C}_j^2 < 0$, the corresponding tensionless string limit lies at a finite distance wall with finitely many generators in its vicinity. This implies that the BPS cone is locally polyhedral, and $\mathcal{C}_j$ is connected to other vertices, where each vertex shares the same face with its neighbors.

[7]More formally, we consider the projectivization of the BPS cone, restrict to one side of the hyperplane $H$ and connect two vertices $\mathcal{C}_j$ to $\mathcal{C}_i$ by boundary edges in this convex set, which is always possible for locally polyhedral spaces.

The first condition for $\mathcal{C}_k$ requires $\mathcal{C}_k \cdot \mathcal{C}_j < 0$, while the latter implies $\mathcal{C}_k \cdot \mathcal{C}_j \geq 0$, since two generators on the same face of the BPS cone can shrink simultaneously and thus belong to the same SCFT or LST. This is a contradiction. Therefore, we conclude that the geometry of the BPS cone requires $\mathcal{C}_i \cdot \mathcal{C}_j \geq 0$ for any pair of distinct generators with $\mathcal{C}_j^2 < 0$.

We now give another argument using the worldsheet theory of BPS strings as well as the effective field theory on 6d spacetime. First note that when both generators $\mathcal{C}_i$ and $\mathcal{C}_j$ (or their associated tensor multiplets) support gauge algebras, this condition follows from the anomaly cancellation condition for $b_i \cdot b_j$, which counts the hypermultiplets charged under both gauge algebras and is therefore positive. Thus, we only need to show that this relation holds when $\mathcal{C}_i, \mathcal{C}_j$ or both do not support a gauge algebra, which occurs only for generators with self-intersections of $-2$, $-1$, or 0.

A string with charge $Q$ and $Q^2 = -2$ that has no associated gauge algebra is referred to as an M-string [47]. This charge $Q$ for an M-string has a unique feature: There is a $\mathbb{Z}_2$ gauge symmetry given by a Weyl reflection when $J \cdot Q = 0$[8] given by the action on charge lattices $C \to C + (C \cdot Q)Q$. Using this gauge symmetry for M-strings we can extend the tensor moduli space beyond the point where $J \cdot Q = 0$. Thus, in the extended tensor moduli space, which covers tensor moduli spaces before and after the Weyl reflection, we can exclude this charge from being a generator. Thus we will focus only on generators with norm $-1$ and 0.

Consider a $\mathcal{C}_i$ with self-intersection $-1$ and no associated gauge algebra, which is an E-string discussed in the previous section, and let $\mathcal{C}_j$ have $\mathcal{C}_j^2 \leq -3$. In this case, $\mathcal{C}_j$ always supports a gauge algebra, and when they intersect, the worldsheet CFT on the E-string has a current algebra for the gauge symmetry on $\mathcal{C}_j$ with level $k = \mathcal{C}_i \cdot \mathcal{C}_j$. Since the E-string has only left-moving degrees of freedom, meaning $c_R = 0$, this level $k$ must be positive. This shows that $\mathcal{C}_i \cdot \mathcal{C}_j \geq 0$ when $\mathcal{C}_i$ is an E-string charge and $\mathcal{C}_j^2 \leq -3$. Additionally, when both $\mathcal{C}_i, \mathcal{C}_j$ are charges for E-strings, such that $\mathcal{C}_i^2 = \mathcal{C}_j^2 = -1$, the central charge formula (2.9) for supergravity strings holds for these two individual strings and also for any possible bound state. This is because an enhanced $SU(2)_I$ R-symmetry that would invalidate formula (2.9) only arises for shrinkable strings; however, we assume $\mathcal{C}_i$ and $\mathcal{C}_j$ cannot shrink simultaneously, so their bound states are also not shrinkable. Also, as explained earlier, an E-string worldsheet theory has no $SU(2)_I$ symmetry. Thus, the central charge formula for supergravity strings applies to two E-strings both before and after they form a bound state in this case. Then, since bound states can only have a larger central charge than the sum of the central charges of two individual strings, meaning $c_R(\mathcal{C}_i + \mathcal{C}_j) \geq c_R(\mathcal{C}_i) + c_R(\mathcal{C}_j)$, the central charge formula for $c_R$ in (2.9) implies that $\mathcal{C}_i \cdot \mathcal{C}_j \geq 0$ when both charges are those of E-strings.

Then, the remaining cases are those where one or both generators have self-intersection 0. Since these null charges are generators in this case, each has a tensionless limit at an infinite distance. As we will show in Section 3.3, at such an infinite distance, the moduli vector $J$ aligns along the direction of the tensionless null generator. The positivity of BPS

---

[8]This is also evident from the fact that the ground state of an M-string, when compactified on a circle, becomes a 5d vector multiplet.

string tensions demands $J \cdot \mathcal{C}_i \geq 0$ for all generators at the asymptotic limit, and thus this leads to the conclusion that all generators must intersect with null generators non-negatively. This completes the proof that any two generators of the BPS cone intersect non-negatively, thereby proving the relation (3.2) for all cases.

Let us now present the detailed proofs for the main claims presented at the beginning of this section. We start with proving that any 6d (1,0) supergravity theory with $T \geq 1$ must have an $H$-string with charge $f$ satisfying (5.1) except possibly for $T = 9$ and $b_0 = 0$.

## 3.2 Blowdowns and the presence of $H$-strings

To establish the presence of an $H$-string, we rely on the fact that the $b_0$ for the gravitational instanton is part of the BPS cone, which implies that the BPS cone must include at least one generator $\mathcal{C}_i$ with $b_0 \cdot \mathcal{C}_i > 0$ (except $b_0 = 0$ cases which we will discuss separately). The only possible generators with this property have self-intersections of either $-1$ or $0$, where the latter corresponds to the $H$-string. Our strategy is to remove all such '$-1$' generators while preserving these properties, which leads to the existence of an $H$-string at the end of the process.

With this strategy in mind, let us first define the concept of a '*blowdown*' procedure for generators $e_i$ with $e_i^2 = -1$ and $b_0 \cdot e_i = 1$, which we often refer to as '$-1$' generators.[9] A blowdown is performed by moving to the boundary of the tensor cone $\partial \mathcal{T}$ (which is possible because $e_i$ is assumed to be a generator of the BPS cone) where $J \cdot e_i = 0$, corresponding to the singular CFT fixed point of the tensor multiplet associated with $e_i$. This process defines a linear map from the BPS cone in $T$-dimensional tensor charge lattice $\Gamma$ to a new (smaller) BPS cone in $(T\text{-}1)$-dimensional lattice $\Gamma'$ where $\Gamma' \subset \Gamma$ is orthogonal to $e_i$. In this process, each generator $\mathcal{C}_j \neq e_i$ is mapped to a new charge $\mathcal{C}'_j$ which lies in the new BPS cone for $\Gamma'$, as follows:

$$\mathcal{C}_j \quad \rightarrow \quad \mathcal{C}'_j = \mathcal{C}_j + (\mathcal{C}_j \cdot e_i)e_i . \tag{3.3}$$

This guarantees that every primed charge is orthogonal to $e_i$, meaning $Q' \cdot e_i = 0$. The self-intersection of the new charge increases or stays the same, as given by

$$\mathcal{C}'^2_j = \mathcal{C}_j^2 + \mathcal{C}_j \cdot e_i(\mathcal{C}_j \cdot e_i - 1) , \tag{3.4}$$

with $\mathcal{C}_j \cdot e_i \geq 0$. At each step in this process, the BPS charge $b_0$ is mapped to

$$b'_0 = b_0 + (b_0 \cdot e_i)e_i = b_0 + e_i \qquad \text{with} \quad b'^2_0 = 9 - (T - 1) . \tag{3.5}$$

One also finds that the level $k_L$ either increases or remains unchanged.

$$k'_L = \frac{1}{2} \left( \mathcal{C}'^2_j - b'_0 \cdot \mathcal{C}'_j + 2 \right) = k_L + \mathcal{C}_j \cdot e_i(\mathcal{C}_j \cdot e_i - 1) \geq k_L . \tag{3.6}$$

Since the positivity conditions $J \cdot \mathcal{C}_j \geq 0$, $J \cdot e_i \geq 0$ and $\mathcal{C}_j \cdot e_i \geq 0$ are satisfied prior to the blowdown process, it is evident that the strings of charge $\mathcal{C}'_j$ will also satisfy $J \cdot \mathcal{C}'_j \geq 0$

---

[9]Another type of generator can also exist, characterized by $\mathcal{C}_i^2 = -1$ and $b_0 \cdot \mathcal{C}_i = -1$. For clarity and to avoid confusion, we will label these as '$-\hat{1}$' generators when needed.

during the process. Therefore, these strings continue to be BPS strings with positive tensions in the tensor moduli space slice, now described as a sub-cone of the original tensor cone $\mathcal{T}$, restricted by $J \cdot e_i = 0$, i.e. $\mathcal{T}' = \mathcal{T}|_{J \cdot e = 0} \subset \mathcal{T}$, after the blowdown. Importantly, those strings with $\mathcal{C}_j'^2 \leq 0$ and $k_L' = 0, 1$ can still reach a tensionless limit at $J \cdot \mathcal{C}_j' = 0$, and serve as generators of the $(T\text{-}1)$-dimensional BPS cone. The sub-cone $\mathcal{T}'$ is the dual cone to the reduced BPS cone generated by these $\mathcal{C}_j'$ strings. Meanwhile, those with $\mathcal{C}_j'^2 > 0$ or $k_L' \neq 0, 1$ are excluded from being generators.

We should clarify that the blowdown procedure we have defined differs from the small instanton transition, which induces a smooth transition from a UV theory with $T$ tensor multiplets to an IR theory with $T$-1 tensor multiplets by Higgsing one tensor multiplet into 29 hypermultiplets, which is discussed in [48–50]. Our blowdown process does not trigger such a transition. Instead, it represents a movement within the tensor moduli space toward its boundary at $J \cdot e_i = 0$, where a CFT fixed point with tensionless strings in the charge class $e_i$ emerges. This is analogous to staying at the CFT point and not moving away from it by giving vacuum expectation values to hypermultiplets.

We will proceed to blow down all the generators $e_i$ in the BPS cone in some order until no more '$-1$' generators with $\mathcal{C}_j^2 = -1$ and $b_0 \cdot \mathcal{C}_j = 1$ remain (including the ones which may have been generated during blowdown), and continue it until $T' = T - \Delta_T \geq 1$, where $\Delta_T$ represents the number of $e_i$'s we have blown down. As we have already noted, at each step of the blowdown where $\mathcal{C}_j \to \mathcal{C}_j'$, the lattice vectors $\mathcal{C}_j'$ *also belongs to the BPS cone before the blowdown*, because $\mathcal{C}_j \cdot e_i \geq 0$ and $\mathcal{C}_j, e_i$ are part of the BPS generators before the blowdown. From (3.3) $\mathcal{C}_j'$ is given by a positive combination of generators and thus belongs to the BPS cone before the blowdown, though it may not necessarily be one of the generators before the blowdown. Moreover, $\mathcal{C}_j' \cdot b_0' = \mathcal{C}_j' \cdot (b_0 + (b_0 \cdot e_i)e_i) = \mathcal{C}_j' \cdot b_0$. This will be important in our argument below, as we establish the existence of an $H$-string (with $f^2 = 0, f \cdot b_0 = 2$) in the blowndown lattice (with $f^2 = 0, f \cdot b_0' = 2$), and argue that it is part of the BPS strings even before the blowdown with the same inner product with $b_0$.

The charge $b_0'$ induced from $b_0$ at the end will satisfy $b_0'^2 = 9 - T'$ and

$$
b_0' \cdot \mathcal{C}_j' = \begin{cases} 2 & \text{if } \mathcal{C}_j'^2 = 0, \; k_L(\mathcal{C}_j') = 0 \\ 1 & \text{if } \mathcal{C}_j'^2 = -1, \; k_L(\mathcal{C}_j') = 0 \\ 0 & \text{if } \mathcal{C}_j'^2 = -2, \; k_L(\mathcal{C}_j') = 0 \; \text{ or } \; \mathcal{C}_j'^2 = 0, \; k_L(\mathcal{C}_j') = 1 \\ \leq -1 & \text{otherwise} \end{cases} \tag{3.7}
$$

for all remaining generators $\mathcal{C}_j'$. This relation arises from the inequality $k_L' \geq k_L \geq 0$ for the generators at each step. As discussed above, $b_0'$ is effective in the reduced BPS cone and satisfies $J \cdot b_0' \geq 0$. Therefore, it must be expressed as a non-negative sum of the generators, given by $b_0' = \sum_i n_i \mathcal{C}_i'$, with integral coefficients $n_i \geq 0$.

As we perform a *sequential* blowdown process, the generators of the BPS strings prior to the blowdown will remain part of the BPS string spectrum afterward, although some may no longer serve as BPS generators. Among the new set of generators we continue to sequentially blowdown the $e_i$ strings, until none is left. This is always possible by the assumption that the tensor cone is dual to the BPS cone. At the end of this process, we will arrive at a reduced BPS cone of dimension $T'$. If a single $e_i$ generator remains, we will

stop at $T' = 1$. We will then consider two possibilities: $T' > 1$ or $T' = 1$. The following flowchart summarizes the procedure:

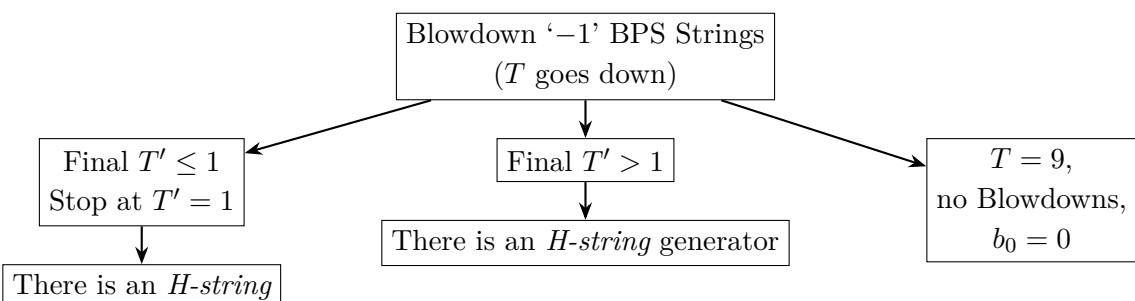

If $T' > 1$, this cone is generated by BPS strings with $\mathcal{C}_j'^2 \neq -1$, since all '$-1$' generators have been blown down. In this case, because there is no remaining '$-1$' generator, to satisfy the condition $J \cdot b_0' \geq 0$, and given that $J$ is a positive linear combination of the remaining BPS strings, either (a) there must be at least one null BPS string $f$ such that $f^2 = 0$ and $b_0' \cdot f = 2$, or (b) all generators must have $\mathcal{C}_j'^2 = -2$ or $0$ with $b_0' \cdot \mathcal{C}_j' = 0$. In the first case, the BPS string $f$ corresponds to the $H$-string we are aiming to identify.

In the case (b), when $b_0' \cdot \mathcal{C}_j' = 0$ for all $j$, the condition $b_0' \cdot Q' = 0$ holds for any effective charge $Q'$ in the reduced BPS cone. This leads to the conclusion that $b_0' = 0$. However, since $b_0'^2 = 9 - T'$, this can only occur when $T' = 9$ (or $T = 9$ theory which does not come from a blowdown, and has no '$-1$' tensor multiplets). When $T' = 9$, we can then consider a blowup to $T' = 10$, where it came from, which reverses the blowdown of a '$-1$' generator $e$. Then $b_0$ would be mapped to a charge $\tilde{b}_0 = -(\tilde{b}_0 \cdot e)e$ at $T' = 10$. However, because $J \cdot e \geq 0$ and $\tilde{b}_0 \cdot e \geq 0$, this charge $\tilde{b}_0$ cannot satisfy the condition $J \cdot \tilde{b}_0 \geq 0$, contradicting its effectiveness. Therefore, a theory with $b_0 \cdot \mathcal{C}_j' = 0$ for all generators can only exist when $T = 9$ and does not come from blowdowns, and it can contain only generators of $\mathcal{C}_j^2 = -2$ or $0$ with $b_0 \cdot \mathcal{C}_j = 0$. This is an isolated theory at $T = 9$, unrelated to any other theories with $T \neq 9$ through the blowdown process. Indeed, such a theory can be engineered by F-theory on the Enriques surface. This theory has an even unimodular charge lattice, which is consistent with the fact that all generators have $\mathcal{C}_j^2 = -2$ or $0$. We will discuss this case further in the next section.

Lastly, let us consider the cases with $T' = 1$. If no '$-1$' generator remains after the blowdowns, we can use the same argument as in the case of $T' > 1$ to prove that the theory must possess an $H$-string with charge $f$. If there are left-over '$-1$' generators with $T' = 1$, there can be at most one them. This can be seen by showing that if there is an additional '$-1$' generator, they would have to span the BPS lattice, and using the fact that inner product of $b_0'$ with each $-1$ generator is 1, and that $b_0'^2 = 8$, which leads to a contradiction. Now, the string charge lattice $\Gamma'$, after the blowdowns, is an unimodular lattice with a signature of $(1, -1)$. There are only two possible types of such lattices: either the odd lattice with $\Omega = \mathrm{diag}(1, -1)$ and $b_0' = (3, -1)$ or the even lattice with $\Omega = \begin{pmatrix} 0 & 1 \\ 1 & 0 \end{pmatrix}$ and $b_0' = (2, 2)$ [23].[10]

<hr />

[10]There is another option for an even lattice with $\Omega = \begin{pmatrix} 0 & 1 \\ 1 & 0 \end{pmatrix}$ and $b_0' = (1, 4)$. In this case, $b_0' \cdot \mathcal{C}_i \geq 0$ is

If no '−1' generator remains after the blowdowns, we can use the same argument as in the case of $T' > 1$ to prove that the theory must possess a rational null BPS string with charge $f$. When we have a '−1' generator, the BPS cone is in an odd lattice, and we can choose $b'_0 = (3, -1)$ and the basis for the '−1' generator as $\hat{e} = (0, 1)$. The other generator can generally take the form $C' = (a, -b)$, where $a, b$ are non-negative integers. This generator must satisfy the conditions $C'^2 \leq 0$ and $k'_L = 0$ or 1. The only solution to these constraints is $a = b = 1$, which corresponds to the null charge $f = (1, -1)$, $b'_0 \cdot f = 2$. This is the effective $H$-string class we were looking for.

Therefore, we conclude that *every consistent 6d supergravity theory with $T > 0$ must include an H-string $f$ in its BPS cone*, except for the isolated $T = 9$ and $b_0 = 0$ cases. In the following sections, the charge $f$ and the associated $H$-string always refer to those we identified here.

## 3.3 SCFTs contained in LST

The second step in our proof is to demonstrate that every string of charge $\mathcal{C}_i$ with a gauge algebra $\mathfrak{g}_i \neq \mathfrak{g}_{\text{small}}$, corresponding to a CFT string, that does not intersect the null charge $f$ must be an element of an LST associated with the charge $f$. Specifically, we will show that

$$\text{If} \quad \mathcal{C}_i \cdot f = 0 \quad \text{and} \quad \mathfrak{g}_i \neq \mathfrak{g}_{\text{small}} , \quad \text{then} \quad \mathcal{C}_i \in \text{LST}(f) , \tag{3.8}$$

for $f^2 = 0$ and $b_0 \cdot f = 2$, where $\mathcal{C}_i$ denotes a generator supporting a gauge algebra $\mathfrak{g}_i$, and $\mathfrak{g}_{\text{small}} \in \{\varnothing, \mathfrak{su}_{2,3,4}, \mathfrak{sp}_2, \mathfrak{g}_2\}$. Here, $\text{LST}(f)$ refers to the BPS cone of a little string theory whose little string carries charge $f$. In other words, we assert that a generator $\mathcal{C}_i$ satisfying $\mathcal{C}_i \cdot f = 0$ and $\mathfrak{g}_i \neq \mathfrak{g}_{\text{small}}$ is also a generator of BPS strings in a little string theory associated with the charge class $f$. In particular, a generator $\mathcal{C}_i^2 \leq -4$ always supports a gauge algebra $\mathfrak{g}_i$ that is not $\mathfrak{g}_{\text{small}}$, and thus, it is a component of an LST in the charge class $f$ when $\mathcal{C}_i \cdot f = 0$. See table 4 for a summary of the allowed gauge algebras and charged matter fields supported on generators. This reveals a non-trivial containment relation between local SCFTs and LSTs embedded in a 6d supergravity. The proof is as follows.

We have shown that the $H$-string with charge $f$, which we identified earlier, will eventually become a generator in $T'$-dimensional BPS cone after sufficient blowdowns, and there it lies at a boundary of the BPS cone. This shows that it must be possible to reach a boundary of the original tensor cone $\mathcal{T}$ where the BPS string in the charge class $f$ becomes tensionless, such that $T_f \sim J \cdot f \to 0$. However, since $f^2 = 0$, this string is not a CFT string, and only CFT strings can become tensionless at finite distances. Therefore, we conclude that every 6d supergravity with $T > 0$ (except cases where $T = 9$ and $b_0 = 0$) must exhibit a tensionless limit for the $H$-string, and this limit must be located at an infinite distance in its tensor moduli space.

At this tensionless limit, while keeping $J^2 = 1$, the moduli vector should align along a null vector, which can be shown as follows. The moduli vector can be expressed as

$$J = cJ_0 + \sum_i a_i \mathcal{C}_i , \qquad \text{as} \quad c \to \infty \tag{3.9}$$

satisfied only when $\mathcal{C}_i = f = (0, 2)$, with $f^2 = 0$ and $b'_0 \cdot f = 2$. Then, one can easily check that there is no other generator that can non-negatively intersect with $f$. Thus, this option is ruled out.

with some finite limiting coefficients $a_i$, where $J_0$ is the dominant class at infinite distance. As shown in [39], the condition $J^2 = 1$ leads to the conclusion that $J_0^2 = 0$ and

$$a_i \sim \begin{cases} \mathcal{O}(1/c) & \text{for} \quad J_0 \cdot \mathcal{C}_i > 0 \\ \mathcal{O}(1) & \text{for} \quad J_0 \cdot \mathcal{C}_i = 0 \end{cases}, \tag{3.10}$$

in the limit. Hence, $J \approx cJ_0 + \mathcal{O}(1)$ with $J_0^2 = 0$ and $c \to \infty$.

In a Lorentzian space with signature $(1, T)$, two null vectors $f$ and $J_0$ must either be proportional or intersect. If $f$ and $J_0$ are not proportional, they must intersect $f \cdot J_0 \neq 0$, which would cause $J \cdot f \gg \mathcal{O}(1)$ in the limit. This contradicts the assumption of a tensionless limit for $f$. Hence, we conclude that $J_0 = f$ and $J \approx cf + \mathcal{O}(1)$ at the infinite distance limit where the tension of the $f$-string vanishes. Moreover, because $b_0 \cdot f = 2$ and $J \cdot b_0 \geq 0$, the coefficient $c$ in the limit must be positive.

Let us now demonstrate that the string tension of a generator, say $\mathcal{C}_i^\perp$, satisfying $\mathcal{C}_i^\perp \cdot f = 0$ also approaches zero in the tensionless limit for the $f$-string. To show this, we write the moduli vector in the infinite distance limit as

$$J = cf + c^{-1}J_+ + \alpha J_\perp , \quad J_+ = \sum_i a_i \mathcal{C}_i^+ , \quad J_\perp = \sum_j b_j \mathcal{C}_j^\perp , \tag{3.11}$$

with some $\mathcal{O}(1)$ coefficients $\alpha, a_i, b_j \geq 0$, and $c \to +\infty$, where $f \cdot \mathcal{C}_i^+ > 0$ and $f \cdot \mathcal{C}_j^\perp = 0$. Here, the condition $f \cdot \mathcal{C}_j^\perp = 0$, and thus $f \cdot J_\perp = 0$, implies that $J_\perp$ is a space-like vector and $J_\perp^2 < 0$. Additionally, since $J \cdot \mathcal{C}_j^\perp \geq 0$ for every $j$, we must have $J \cdot J_\perp \geq 0$. Consequently, $J_\perp$ must intersect with $J_+$, and the coefficient $\alpha$ should scale as $\alpha \sim c^{-1}$, while $a_i, b_j$ remain $\mathcal{O}(1)$, in the limit $c \to \infty$. Therefore, we conclude that the tension of strings with charge $\mathcal{C}_j^\perp$ vanishes in this limit, i.e., $J \cdot \mathcal{C}_j^\perp \sim \mathcal{O}(c^{-1}) \to 0$ as $c \to \infty$. So the tensions of both $f$ and $\mathcal{C}_i^\perp$ vanish at the same rate of $\mathcal{O}(c^{-1})$ in this asymptotic limit.

We now show that $J \cdot \mathcal{C}_j^\perp / J \cdot f$ is bounded above in the tensor moduli space. Since this is a continuous function on the tensor moduli space, its boundedness at all infinite distance limits ensures it is bounded throughout the entire tensor moduli space. We have already shown that this function is bounded at the infinite distance limit corresponding to $f$. If there is another null charge $f'$ with $f'^2 = 0$, we may have another infinite distance limit. In this case, $f'$ must intersect with $f$, which causes the tension of the $f$-string to diverge linearly in $c$ in the infinite distance limit for $f'$. Meanwhile, the tension of the string associated with $\mathcal{C}_i^\perp$ will either vanish if $f' \cdot \mathcal{C}_i^\perp = 0$ or diverge linearly in $c$, the same rate (up to scaling) as the $f$-string. Thus, as an $f$-string shrinks, the $\mathcal{C}_i^\perp$-string also shrinks at the same rate, and when the tension of the $\mathcal{C}_i^\perp$-string diverges, the $f$-string tension also diverges at the same rate, but not vice versa. This shows that $J \cdot \mathcal{C}_j^\perp / J \cdot f$ is bounded above at every infinite distance limit. Thus, if the total number of null charges, including both $f$ and all other $f'$, in the BPS cone is finite, we can establish the following relation throughout the entire tensor moduli space:

$$mJ \cdot f \geq nJ \cdot \mathcal{C}_i^\perp , \tag{3.12}$$

with some 'finite' positive integers $m$ and $n$ for any generator $\mathcal{C}_i^\perp$ with $f \cdot \mathcal{C}_i^\perp = 0$.

However, it may be possible that the number of null charges in the BPS cone is infinite. This can occur only when the cone is non-polyhedral and generated by infinitely many BPS charges. In such a case, our argument may fail, as $f' \cdot C_i^\perp \to \infty$ could diverge with infinitely many $f'$. Fortunately, we can show that $Q \cdot C_i$ is finite for every null charge $Q$ if the gauge algebra $\mathfrak{g}_i$ supported on a generator $C_i$ is not of type $\mathfrak{g}_{\text{small}}$. Note that $Q$ here is a null charge and the associated BPS strings are either $H$-string ($c_L = 20$) or Type II string ($c_L = 4$). When $Q \cdot C_i = k_i > 0$, the gauge algebra $\mathfrak{g}_i$ induces a current algebra at level $k_i$, but its contribution $c_{\mathfrak{g}_i}$ to the left-moving central charge given in (2.13) cannot be bigger than 20 or 4. This means that if $\mathfrak{g}_i$ has a dimension bigger than 20, hence not of type $\mathfrak{g}_{\text{small}}$, then the level or the intersection number $k_i$ is bounded and finite. Therefore, if the gauge algebra on a generator $C_i$ is not of type $\mathfrak{g}_{\text{small}}$, we can apply the same reasoning to establish the relation (3.12) in cases $C_i \cdot f = 0$, regardless of whether the number of generators is finite or infinite.

Thus, we have proven that for any generator, say $\hat{C}_i$, satisfying $C_i \cdot f = 0$ and $\mathfrak{g}_i \neq \mathfrak{g}_{\text{small}}$, the following holds:

$$J \cdot (mf - n\hat{C}_i) \geq 0 \ , \tag{3.13}$$

for any $J$ in the tensor cone. All generators with $C_i^2 \leq 4$ and $C_i \cdot f = 0$ are in this class. This leads to the conclusion that $mf - n\hat{C}_i$ lies inside the BPS cone, though it may only represent a lattice element and not necessarily correspond to a BPS charge.

Consider the set of all lattice elements inside the BPS cone, even if they do not correspond to a BPS string. Since the BPS subspace should have finite index inside the full lattice (as over reals it spans the full space), there must exist a finite integer multiple of $mf - n\hat{C}_i$, say $p(mf - n\hat{C}_i)$ with $p > 0$, that is a BPS string. Now, the positive sum of two BPS charges as

$$p \times (mf - n\hat{C}_i) + pn\hat{C}_i = pmf \ , \tag{3.14}$$

in turn implies that, since an integer multiple of $f$ can be decomposed into BPS strings charges, each of these belongs to the BPS cone $\text{LST}(f)$. Moreover, from the classification of LSTs, we know that if a multiple of any generator belongs to an LST BPS cone, then the generator itself must also be part of that same cone. Thus, we conclude that *any generator $C_i$ with $f \cdot C_i = 0$ and $\mathfrak{g}_i \neq \mathfrak{g}_{\text{small}}$ in a supergravity theory is a generator of the BPS cone for an LST corresponding to the charge class $f$.*

This conclusion is also consistent with the known fact from LST classification that any LST can be constructed by gluing a collection of SCFT tensor multiplets (or atoms), and that the little string charge $f$ in the LST is formed by a sum of the charges $C_i$ of the SCFT strings satisfying $C_i^2 < 0$ and $f \cdot C_i = 0$.

It is also worth mentioning that several LSTs can exist in the same charge class $f$. While these LSTs are independent, they can all share the same charge $f$ and simultaneously shrink in the tensionless limit for $f$. A generator $C_i$ with $f \cdot C_i = 0$ will be an element of one of these LSTs.

| $\mathfrak{g}_i$ | $H_i$ | $b_i^2$ | $b_0 \cdot b_i$ | $\Delta_i = H_i - V_i$ |
|---|---|---|---|---|
| $\mathfrak{e}_8$ | | $-12$ | $-10$ | $-248$ |
| $\mathfrak{e}_7$ | $\frac{n}{2} \times \mathbf{56}$ $(n \leq 3)$ | $-8+n$ | $-6+n$ | $-133+28n$ |
| $\mathfrak{e}_6$ | $n \times \mathbf{27}$ $(n \leq 1)$ | $-6+n$ | $-4+n$ | $-78+27n$ |
| $\mathfrak{f}_4$ | | $-5$ | $-3$ | $-52$ |

**Table 2**. Gauge algebras with $\Delta_i + 29 \leq 0$.

## 3.4 Bound on Tensor multiplets

Lastly, we will prove that the number of tensor multiplets in 6d supergravity cannot be arbitrarily large, utilizing the classification of 6d SCFTs and LSTs, as well as the unitarity of the 2d (0,4) CFT associated with the $H$-string in the charge class $f$. In this section we use a relatively simple argument to establish this. In the following section with more effort we find a sharp bound for $T$.

Let us begin by briefly discussing how infinite families of 6d supergravity theories with arbitrarily large values of $T$ can arise, as explained in detail in [10, 31, 34]. The gravitational anomaly cancellation condition, which is the first equation in (2.7), imposes a strict constraint on the number of tensor multiplets and the types of gauge algebras. The left-hand side of the condition (excluding neutral hypermultiplets) is bounded by 273. As analyzed in [10], only specific gauge algebras, listed in Table 2, contribute negatively to the gravitational anomaly. These algebras come with a tensor multiplet having self-intersection $-n$ in the range $5 \leq n \leq 12$, and when embedded in supergravity, their tensor charges become generators of the BPS cone. This also implies that as we increase the number of these gauge algebras, the number of tensor multiplets also grows, contributing $\Delta_{G_i} + 29$ to the anomaly, where 29 accounts for the contribution of a single tensor multiplet. Since $\Delta_{G_i} + 29$ is negative for all these gauge algebras in Table 2, it is possible, in principle, to have an arbitrarily large number of tensor multiplets without violating the gravitational anomaly bound.

On the other hand, all other gauge algebras, which include a tensor contribution 29 when $b_i^2 < 0$ but might not include an additional tensor multiplet when $b_i^2 \geq 0$, contribute positively to the gravitational anomaly. Thus, once the number of gauge algebras listed in Table 2 is fixed, the gravitational anomaly imposes an upper limit on the number of other gauge algebras. Hence, to prove that the number of tensor multiplets is bounded, it is enough to show that a consistent supergravity theory can only accommodate a finite number of gauge algebras from the types listed in Table 2.

We will decompose the generators, say $\mathcal{D}_i$, associated with the tensor node supporting a gauge algebra $\mathfrak{g}_i$ of the types in Table 2, into two classes: those with $\mathcal{D}_i \cdot f > 0$ and those with $\mathcal{D}_i \cdot f = 0$, where $f$ is the charge class of the *H-string* we identified earlier. We will first prove that the number of generators with $\mathcal{D}_i \cdot f > 0$ is finite. This can be easily demonstrated using the unitarity of a $H$-string with charge $f$. The left-moving central charge for this string is $c_L = 20$ as given in (2.15). As discussed in Section 2.3, when $\mathcal{D}_i \cdot f > 0$, the worldsheet degrees of freedom on the $f$-string must realize a current algebra

for the gauge algebra $\mathfrak{g}_i$ corresponding to $\mathcal{D}_i$. The total rank of the gauge algebra $\prod_i \mathfrak{g}_i$ for all $\mathcal{D}_i$'s with $\mathcal{D}_i \cdot f > 0$ cannot exceed the left-moving central charge $c_L = 20$. As a result, the rank of the gauge algebra $\prod_i \mathfrak{g}_i$ cannot be bigger than 20, and thus the number of generators (or tensor multiplets) $\mathcal{D}_i$ with $\mathcal{D}_i \cdot f > 0$ is bounded.

Next, we need to show that the number of generators $\mathcal{D}_i$ with $\mathcal{D}_i \cdot f = 0$ is also finite. As demonstrated above these generators (and the corresponding SCFTs) must be parts of an LST in the class $f$, as $\mathcal{D}_i^2 \leq -4$ and thus the associated gauge algebras are not of type $\mathfrak{g}_{\text{small}}$. This containment relationship between the SCFTs and the little string theory imposes significant constraints on the gauge algebras and matter content associated with those generators. So let us first analyze the structure of LSTs and the constraints on the generators derived from the SCFT and LST classifications.

For this purpose, we will briefly review the basic structures of SCFTs and LSTs, which have been extensively studied in [13, 15]. Additional details can be found in Appendix A. The base of any SCFT and LST can be constructed by gluing basic building blocks known as "atoms". There are two types of building blocks:

$$\text{DE type}: \quad \overset{\mathfrak{so}_8}{4}, \overset{\mathfrak{e}_6}{6}, \overset{\mathfrak{e}_7'}{7}, \overset{\mathfrak{e}_7}{8}, \overset{\mathfrak{e}_8'''}{9}, \overset{\mathfrak{e}_8''}{10}, \overset{\mathfrak{e}_8'}{11}, \overset{\mathfrak{e}_8}{12} \tag{3.15}$$

$$\text{non-DE type}: \quad \overset{\mathfrak{su}_3}{3}, \overset{\mathfrak{su}_2\mathfrak{g}_2}{2\ 3}, \overset{\mathfrak{su}_2\mathfrak{so}_7\mathfrak{su}_2}{2\ 3\ 2}, \overset{\mathfrak{sp}_1\mathfrak{g}_2}{2\ 2\ 3}, \overset{\mathfrak{f}_4}{5}, \text{ ADE graphs}, \tag{3.16}$$

where the integer $n$ represents the self-intersection $-n$ for each tensor multiplet. Additionally, the minimal gauge algebra that arises after full Higgsing is assigned to each tensor multiplet. The gauge algebras $\mathfrak{e}_7'$ stands for the $\mathfrak{e}_7$ algebra with $\frac{1}{2}\mathbf{56}$ hypermultiplets, while $\mathfrak{e}_8', \mathfrak{e}_8'', \mathfrak{e}_8'''$ correspond to 1, 2, 3 small $\mathfrak{e}_8$ instantons, respectively.

These atoms can be connected to other atoms via a '$-1$' tensor multiplet. The term "DE" indicates that the tensor multiplet can support minimal gauge symmetry of D- or E-type. We also call DE-type atoms as "nodes", denoted by $g_i$, while non-DE-type atoms can combine to form other building blocks called "links", denoted by $L_{i,i+1}$ or $S_i$. For general configurations with more than five nodes, the base takes the shape of a tree diagram, as detailed in Section 5.5 in [13]

$$S_1 \overset{S_1'}{g_1} L_{1,2} \overset{\mathbf{I}^{\oplus s}}{g_2} L_{2,3} g_3 ... L_{m-1,m} g_m L_{m,m+1} ... L_{k-2,k-1} \overset{\mathbf{I}^{\oplus t}}{g_{k-1}} L_{k-1,k} \overset{\mathbf{I}^{\oplus u}}{g_k} S_k . \tag{3.17}$$

Here, the interior link $L_{i,i+1}$ connects the two nodes $g_i$ and $g_{i+1}$, while $S_1, S_1', S_k$ represent side links attached to the end nodes $g_1$ or $g_k$. The instanton side link $\mathbf{I}^{\oplus m}$ corresponds to $m$ small instantons of the form $\underbrace{122\cdots2}_{m}$. When there are fewer than six nodes, the base structure takes on a specific configuration that depends on the number of nodes. Once the base is established, appropriate gauge algebras can be chosen for each tensor multiplet in a manner that fulfills the anomaly cancellation requirements.

It has been explored in [15] that any LST can be derived from an appropriate SCFT, whose base structure takes the form of (3.17) for more than five nodes, by adding an additional tensor multiplet. Conversely, removing a tensor multiplet from an LST results in an SCFT. The rules for extending SCFTs into LSTs are thoroughly studied in [15]. See also

[14, 17] for related discussions. As a result, every LST base with more than five nodes takes the form of (3.17) with an additional tensor multiplet such that the tensor intersection form $\Omega_{\mathrm{LST}}$ is negative semi-definite with a null direction. Our proof relies solely on LSTs associated with an *H*-string. This means the corresponding LST endpoint is classified as '*P*' according to [15]. Therefore, the topology of an LST base we are interested in is always tree-shaped, excluding any loop-like configurations.

Each building block of the base in (3.17) consists of a finite number of tensor multiplets. Additionally, two nodes must be connected by an interior link, and side links can only be attached to the two leftmost and rightmost nodes when there are more than five nodes. This implies that if the number of interior links is finite, the total number of tensor multiplets in an LST is also finite. We will now demonstrate that the number of interior links in an LST embedded in 6d supergravity is finite.

An interior link connects two DE-type nodes, and all possible internal links are listed in Appendix D of [13]. We focus on the links connecting one or two E-type nodes, as an infinite number of such links is necessary for a theory with infinitely many tensor multiplets. Notably, the presence of frozen singularities does not affect the structure of these interior links. Therefore, our proof in the subsequent discussion in this section holds for all LSTs in the classification, without modification, even in the presence of frozen singularities.

We listed all possible interior links in (A.1). For each interior link, the gauge symmetries on the adjacent nodes are all gauged. Also, the gauge node $g_i$, for $2 < i < k - 2$, is shared by only two links, $L_{i-1,i}$ and $L_{i,i+1}$. This allows us to calculate gravitational anomaly contribution from each interior link as follows. First, we compute the anomaly contributions from the tensor, hyper, and vector multiplets contained in the interior link. When two interior links are connected by a node with a DE-type gauge algebra, we also include the anomaly contribution from the DE-type gauge algebra and one tensor multiplet for the connecting node. Half of this additional anomaly contribution is assigned to each link connected by the node. Thus, for every interior link, the total anomaly contribution we will calculate below is the sum of the anomaly contributions from the matter content within the link and half of the anomaly contributions from the adjacent nodes. When calculating this for a base with multiplet interior links, the anomaly contributions from the interior nodes are excluded to prevent double counting.

For example, a link $\mathfrak{e}_7 \overset{3,3}{\otimes} \mathfrak{e}_7$ consists of 5 tensor multiplets with $\mathfrak{su}_2 \times \mathfrak{so}_7 \times \mathfrak{su}_2$ gauge algebra and 8 hypermultiplets for each $\mathfrak{su}_2 \times \mathfrak{so}_7$ pair. The total gravitational anomaly contribution from this matter content is

$$E_7 \overset{3,3}{\otimes} E_7 \quad \rightarrow \quad H + 29T - V = 16 + 5 \times 29 - (21 + 3 + 3) = 134 \ , \qquad (3.18)$$

where $21 + 3 + 3$ corresponds to the sum of the dimensions of the gauge algebras. Now, the $\mathfrak{e}_7$ gauge algebras (without charged matters) on two adjacent nodes are gauged and half of their gravitational anomaly contribution is

$$\frac{1}{2}\left((H + 29T - V)_i + (H + 29T - V)_{i+1}\right) = \frac{1}{2}\left((29 - 133) + (29 - 133)\right) = -104 \ , \qquad (3.19)$$

where 133 is the dimension of an $\mathfrak{e}_7$ gauge algebra. Therefore, we compute the total anomaly as

$$\Delta(\mathfrak{e}_7 \overset{3,3}{\otimes} \mathfrak{e}_7) = 134 - 104 = 30 \ , \qquad (3.20)$$

which represents the gravitational anomaly contribution from the interior link $\mathfrak{e}_7 \overset{4,4}{\otimes} \mathfrak{e}_7$, contributing 134, together with half of the contributions from the two adjacent nodes with $\mathfrak{e}_7$ algebra, contributing $-104$. Since this contribution is positive, having arbitrarily many interior links and nodes of this type would eventually violate the gravitational anomaly cancellation condition.

Similarly, we compute the gravitational anomaly contributions from all other interior links. We summarize the results in Appendix A.3 and A.4. Surprisingly, all interior links contribute positively, and each contribution is actually greater than or equal to 27. Moreover, as discussed in [13], infinitely many interior links can exist only if they are of the minimal type. On the other hand, non-minimal links can only appear as the first or last interior link. Thus, only the links $\mathfrak{e}_6 \overset{2,2}{\otimes} \mathfrak{e}_6$, $\mathfrak{e}_7 \overset{3,3}{\otimes} \mathfrak{e}_7$, and $\mathfrak{e}_8 \overset{5,5}{\otimes} \mathfrak{e}_8$, known as conformal matters in [51, 52], can be placed infinitely many times in the base. The gravitational anomaly contributions from these conformal matter links are all 30. This is naturally expected, as conformal matter arises from placing M5-branes on an ADE-type singularity [51]. Each M5-brane contributes 30 to the gravitational anomaly, which comes from a (2,0) tensor multiplet (consisting of $T = 1, H = 1, V = 0$). Since these conformal matter links represent a single M5-brane on the singularity, their anomaly contribution matches that of a single M5-brane, which is 30.

We can therefore conclude that, since the conformal matter links, together with adjacent nodes, contribute positively to the gravitational anomaly, it is impossible to have infinitely many interior links in an SCFT embedded in 6d supergravity.[11] Consequently, the number of tensor multiplets for the generators $\mathcal{C}_i \cdot f = 0$ embedded in the LSTs in the charge class $f$, corresponding to an $H$-string, is finite. Furthermore, the addition of any generator with $\mathcal{C}_i \cdot f = 0$ that is not part of any LST, which is possible only when $\mathcal{C}_i^2 \geq -3$ and $\mathfrak{g}_i \in \mathfrak{g}_{\text{small}}$, provides a positive contribution to the gravitational anomaly.[12] Combined with the fact that the rank of gauge algebras on tensor multiplets for any charge $Q$ with $Q \cdot f > 0$ is bounded, this proves that *the number of tensor multiplets in a consistent 6d supergravity theory is bounded and finite.*

The finiteness of the number of tensor multiplets also implies that the number of massless fields in 6d supergravity is bounded. This can be understood as follows. First, the $f$-string must intersect with at least one other BPS string generator $\mathcal{C}_i$, since the real BPS cone is $T + 1$ dimensional and the signature of tensor moduli is $(1, T)$. Therefore,

---

[11]This conclusion holds even after involving other generators outside of the SCFT or LST in a supergravity. Note that these conformal matter links and their adjacent nodes lack any remaining hypermultiplets that could be gauged. Thus, only '$-1$' or '$-2$' tensor multiplet without gauge algebra can intersect with these conformal matter links, which increases gravitational anomaly contributions.

[12]The minimum gravitational anomaly contributions from these generators are $29, 30$, and $21$ for '$-1$', '$-2$', '$-3$' generators, respectively, when averaging the contributions of charged hypermultiplets shared between two or more gauge algebras.

at least one tensor multiplet in each LST must intersect with $\mathcal{C}_i$. If the generator with $f \cdot \mathcal{C}_i > 0$ does not support a gauge algebra, the central charge $c_L(\mathcal{C}_i)$ of the string for $\mathcal{C}_i$ will bound the rank of the gauge symmetry on the intersecting tensor multiplets in the LSTs. If the generator with $f \cdot \mathcal{C}_i > 0$ does support a gauge algebra, which has a finite rank, then the mixed-gauge anomaly cancellation will bound the rank of the gauge symmetry on the intersecting tensor multiplets. As a result, the rank of gauge symmetry on at least one tensor multiplet in each LST is bounded, and the same logic applies to adjacent tensor multiplets, propagating across all tensor multiplets in the LST. Now, note that the number of tensors in the LSTs embedded in a supergravity is finite. This therefore implies that the rank for the gauge algebras in the LST is bounded. Meanwhile, the rank of gauge algebras supported on charges $Q$ with $Q \cdot f > 0$ is bounded by 20, as explained earlier. Additionally, since the number of tensor multiplets is finite, the rank of gauge algebras $\mathfrak{g}_i = \mathfrak{g}_{\text{small}}$ on generators with $\mathcal{C}_i \cdot f = 0$ and $\mathcal{C}_i^2 \geq -3$ is also bounded above. This proves that the total rank of gauge algebras is finite, and thus the number of vector multiplets is finite.

Lastly, with finite numbers of vector and tensor multiplets, the gravitational anomaly cancellation imposes a bound on the number of hypermultiplets. Thus, we conclude that the total number of massless fields in a 6d supergravity theory is finite. In the next section, we will determine the precise bounds for the number of tensor multiplets and the rank of gauge algebras, covering both non-Abelian and Abelian ones, in 6d supergravity.

## 4 Precise bounds on the number of Tensors, Vectors and Hypers

To determine the precise bounds on the number of tensor multiplets and gauge algebra ranks, we will further refine the approach used to prove finiteness in the previous subsection. The key to this task is optimizing the structure of LSTs in the charge class $f$ of the $H$-string we identified earlier, ensuring they can support the maximum number of matter fields while maintaining anomaly cancellation conditions. This will therefore involve an in-depth analysis of all possible bases and gauge algebras for LSTs, guided by the classification of 6d SCFTs and LSTs. We will show that LSTs with a large number of matter fields, when embedded in 6d supergravity, are significantly constrained. By analyzing these 'large' LSTs associated with an $H$-string, we will determine the exact bounds. After finding the bound on the number of tensors $T$ we show the rank of the gauge group $r(V)$ as well as $r(V) + T$ are also bounded. Furthermore, we construct explicit supergravity models that saturate these bounds. Finally, we put a bound on the number of hypermultiplets $H$, and neutral hypers $H_0$.

### 4.1 Exact bound on Tensors

Our first task is to solve the optimization problem of maximizing the number of tensor multiplets in a supergravity theory. This starts with examining all possible nodes and links in the classification of 6d SCFTs and LSTs and identifying the LST base structure with a '$P$'-type endpoint that can provide largest number of tensor multiplets. We then analyze potential 'external' tensor multiplets (i.e., not part of the LST) for charges $\mathcal{D}_i$ with $f \cdot \mathcal{D}_i > 0$, which intersect with the 'internal' tensor multiplets in the LST base we

identified. These external tensor multiplets can support gauge algebras with a rank of up to 20, but their gravitational anomaly contributions need to be as negative as possible to enable the base to accommodate more nodes and links. Afterward, we will explicitly construct LSTs, each associated with an *H*-string with charge $f$, with matter fields coupled to these gauge algebras and containing the maximum number of tensor multiplets. As previously discussed, tensor multiplets in LSTs, which are accompanied by other matter fields, contribute positively to the gravitational anomaly, and this bounds their maximum number. Hence, since the classification of SCFTs and LSTs provides all possible matter configurations, the optimization problem can ultimately be solved.

We begin by examining the full list of possible long bases for SCFTs and LSTs, which are classified in Appendix B in [13] and in Appendix C in [15]. These bases share a similar structure: a long chain of identical conformal matters, such as $g \otimes g \otimes \cdots \otimes g$ with $g = \mathfrak{e}_6, \mathfrak{e}_7, \mathfrak{e}_8, \mathfrak{so}_{2n}$. A few interior links may be attached at both ends, and then side links can be attached to the two leftmost and rightmost nodes. Each conformal matter takes the form of

$$\mathfrak{so}_{2n} \otimes \mathfrak{so}_{2n} : \ \mathfrak{so}_{2n} \, 1 \, \mathfrak{so}_{2n} \ , \qquad \mathfrak{e}_6 \otimes \mathfrak{e}_6 : \ \mathfrak{e}_6 12321 \mathfrak{e}_6 \ ,$$
$$\mathfrak{e}_7 \otimes \mathfrak{e}_7 : \ \mathfrak{e}_7 12321 \mathfrak{e}_7 \ , \quad \mathfrak{e}_8 \otimes \mathfrak{e}_8 : \ \mathfrak{e}_8 12231513221 \mathfrak{e}_8 \ . \tag{4.1}$$

As shown in Appendix A, the gravitational anomaly contribution is the same for all these conformal matters, with '$\Delta(g \otimes g) = 30$'. However, the conformal matter $\mathfrak{e}_8 \otimes \mathfrak{e}_8$ contains the largest number of tensor multiplets, with a total of 12, including 11 tensor multiplets from an interior link and two half-tensor multiplets from two adjacent nodes. Furthermore, from the calculations summarized in (A.3) and (A.4), we observe that all other interior links, except for $\mathfrak{e}_8 \otimes \mathfrak{so}$ with $\Delta = 27$ and 6 tensors, have gravitational anomaly contributions $\Delta(g \otimes g) > 30$, but contain fewer tensor multiplets than $\mathfrak{e}_8 \otimes \mathfrak{e}_8$. Also, we found that no side link, which is classified in Appendix D in [13], has a gravitational anomaly contribution less than or equal to 30 while containing more than 11 tensor multiplets. Therefore, the most efficient way to construct a long LST base while maximizing the number of tensor multiplets is to embed as many $\mathfrak{e}_8 \otimes \mathfrak{e}_8$ structures as possible, assuming that all $\mathfrak{e}_8$ symmetries are gauged, and add only the minimum number of tensor multiplets necessary to complete the LST base.

Consider a linear chain consisting of $N$ copies $\mathfrak{e}_8 \otimes \mathfrak{e}_8$ conformal matters. After blowing down all the '$-1$' tensor multiplets in the interior links, the base reduces to

$$12222 \underbrace{322 \cdots 223}_{N-1} 22221 \ . \tag{4.2}$$

To complete this base into an LST with a '*P*'-type endpoint, we need to attach a '$-1$' tensor multiplet to the first and the last nodes, each supporting an $\mathfrak{e}_8$ gauge algebra, which reduces to '$-3$' nodes after the blowdowns. Thus, the LST base that can host the maximum number of tensor multiplets takes the form of

$$[\mathfrak{e}_8] \otimes \underbrace{\overset{1}{\mathfrak{e}_8} \otimes \mathfrak{e}_8 \otimes \cdots \otimes \mathfrak{e}_8 \otimes \overset{1}{\mathfrak{e}_8}}_{N-1} \otimes [\mathfrak{e}_8] \ , \tag{4.3}$$

where $[\mathfrak{e}_8]$ represents the $E_8$ flavor symmetry, whereas the internal $\mathfrak{e}_8$ denotes an $\mathfrak{e}_8$ gauge algebra on a '$-12$' tensor multiplet. The '1' over the first and the last internal $\mathfrak{e}_8$'s are the '$-1$' tensor muliplets we added to complete to the LST base. After subtracting one null direction, this LST base contains a total of $12N$ tensor multiplets with total gravitational anomaly contribution $\Delta(LST) = 30N + 248$ for $N > 2$.[13]

Now, we can embed this LST associated with the $H$-string in a 6d supergravity theory. Note that the LST has an $E_8 \times E_8$ flavor symmetry, which can at least be partially broken or gauged in the gravity theory to introduce additional vector multiplets. These vector multiplets contribute negatively to the gravitational anomaly, specifically $-248 \times 2 = -496$ when fully gauged, and thus enable us to accommodate more tensor multiplets. For this, we can introduce two additional generators $\mathcal{D}_i$ with $f \cdot \mathcal{D}_i > 0$ that support the $E_8 \times E_8$ gauge symmetry. This is allowed because the gauge algebra has a rank of 16, which is below the bound of 20. By attaching two generators with $\mathcal{D}^2 = -12$ to both ends of the LST base, we gauge the two $E_8$ symmetries and complete the tensor multiplet configuration for the supergravity theory.

In principle, we can have two or more LSTs in the same charge class $f$. To optimize the number of tensor multiplets, each LST would follow the same base structure as previously described, but each would contribute to the gravitational anomaly as $\Delta(LST_i) = 30N_i + 248$. Therefore, the number of tensor multiplets is maximized when there is only a single LST in the charge class $f$. Moreover, the charge class $f$ itself cannot support a gauge algebra, because $f \cdot \mathcal{D}_i > 0$ for the generators of the $E_8$ symmetries requires hypermultiplets between $E_8$ and the gauge algebra on $f$, but $E_8$ on $\mathcal{D}_i$ cannot couple to hypermultiplets.

We could also introduce more generators that intersect with this LST, but doing so would only reduce the total number of tensor multiplets in the supergravity theory for the following reasons. These extra generators must intersect with at least one tensor multiplet in the LST, but they cannot intersect with any tensor multiplet that supports a gauge algebra since no flavor symmetry remains. Therefore, they can only intersect with a '$-1$' tensor multiplet, corresponding to an E-string, or a '$-2$' tensor multiplet, corresponding to an M-string. E-strings can host a rank 8 current algebra, and M-strings can host a rank 1 current algebra, but these algebras from the base configuration in (4.3) are already fully occupied. Therefore, these extra generators cannot support any gauge algebra. As a result, adding them, since they contribute 29 for a '$-1$' tensor multiplet or 30 for a '$-2$' tensor multiplet to the gravitational anomaly, would reduce the number of conformal matters in the LST, thereby it only decreases the total number of tensor multiplets. For the same reason, adding more generators supporting gauge algebras $\mathfrak{g}_i = \mathfrak{g}_{\text{small}}$ that neither intersect $f$ nor belong to $f$ would only decrease the total number of tensor multiplets. Hence, adding more of such generators is not allowed for achieving the maximum number of tensor multiplets.

Likewise, we might consider the possibility of introducing additional gauge algebras, including Abelian symmetries, on charges $b_i$ with $b_i^2 \geq 0$, but this is also prohibited. These charges must have $f \cdot b_i > 0$, which means they need to intersect with at least one tensor

---

[13]Naively, we would expect a gravitational anomaly contribution of $30N$ for $N$ copies of the conformal matter. But because the $\mathfrak{e}_8 \times \mathfrak{e}_8$ flavor symmetry at both ends is not yet gauged before embedding the LST into a gravitational theory, there is an additional $+248$ contribution.

multiplet in the LST for the $f$-string. However, as discussed earlier, none of the tensor multiplets in the LST can intersect with $b_i$'s supporting gauge algebras.

Lastly, it may be tempting to consider larger gauge algebras for generators $\mathcal{D}_i$ with $f \cdot \mathcal{D}_i > 0$, as this might provide more negative contributions to the gravitational anomaly and potentially allow for more tensor multiplets. Currently, the $E_8 \times E_8$ symmetry on these generators has a rank of 16, which is below the rank bound of 20. Therefore, one might consider larger gauge algebras with rank 20 on $\mathcal{D}_i$, such as $SO(40)$ or $Sp(20)$. In such cases, a different LST base configuration would be necessary, as the base from (4.3) cannot couple to other gauge algebras. It turns out that these larger gauge algebras can couple with an LST base formed by a long chain of $\mathfrak{so} \otimes \mathfrak{so}$ conformal matters, dressed by side links made only of '$-1$' and '$-2$' tensor multiplets. However, as discussed, this base cannot form a longer chain than that of (4.3) due to gravitational anomaly constraints. Additionally, $\mathfrak{so}$ and $\mathfrak{sp}$ gauge algebras always come with charged hypermultiplets, so the resulting contributions to the gravitational anomaly are not sufficient to host more tensor multiplets. Nevertheless, LST bases composed of '$-1$', '$-2$', and '$-4$' tensor multiplets, though not accommodating many tensor multiplets, can still support a large number of vector multiplets. We will explore these bases further in Section 4.2 when discussing the bounds on gauge algebra ranks.

We thus conclude that the LST base given in (4.3), along with two additional '$-12$' tensor multiplets at both ends, provides a tensor multiplet configuration for 6d supergravity that maximizes the number of tensor multiplets. In total, this supergravity theory will have $12N + 1$ tensor multiplets, with a gravitational anomaly contribution from the charged sector of $\Delta_{\text{tot}} = 30N - 219$. From this, we find that the maximum allowable value of $N$, based on the gravitational anomaly cancellation condition $\Delta_{\text{tot}} \leq 273$, is $N = 16$.

Therefore, the following tensor configuration achieves the maximum number of tensor multiplets in 6d supergravity:

$$\overset{\mathfrak{e}_8}{(12)} \otimes \underbrace{\overset{1}{\mathfrak{e}_8} \otimes \mathfrak{e}_8 \otimes \cdots \otimes \mathfrak{e}_8 \otimes \overset{1}{\mathfrak{e}_8}}_{15} \otimes \overset{\mathfrak{e}_8}{(12)} \; . \tag{4.4}$$

Here, we denote the two '$-12$' tensor multiplets at both ends as $\overset{\mathfrak{e}_8}{(12)}$, meaning that each one gauges an $\mathfrak{e}_8$ flavor symmetry of the LST. This supergravity theory contains in total $T = 193$ tensor multiplets and gauge algebra

$$G = \mathfrak{e}_8^{17} \times \mathfrak{f}_4^{16} \times \mathfrak{g}_2^{32} \times \mathfrak{sp}_1^{32} \; , \tag{4.5}$$

where each $\mathfrak{g}_2 \times \mathfrak{sp}_1$ pair is connected by a bi-fundamental half-hypermultiplet, and every $\mathfrak{sp}_1$ has an extra fundamental half-hypermultiplet. Additionally, there exists 12 neutral hypermultiplets. As explained above, no other gauge algebras can be introduced, and thus no more matter can be added. This is therefore the complete matter content of the supergravity theory with 193 tensor multiplets. Gravitational anomaly cancellation works, with $H = 12 + 256, T = 193, V = 5592$. This establishes the bound $T \leq 193$ on the number of tensor muiltiplets.

Indeed, this theory has a string theory realization, as investigated in [29, 53–55], and it was conjectured in [56] that this represents the maximum number of tensors allowed in F-theory realizations. It corresponds to $E_8 \times E_8$ heterotic string compactified on a singular $K3$ with an $E_8$ singularity, where we place 24 point-like instantons for one of the $E_8$'s on the $E_8$ singularity. Each point-like instanton corresponds to an M5-brane probing the $E_8$ singularity. However, due to curvature terms, there is a shift in M5-brane number by $-8$ leading to $N = 16$ M5-branes probing the $E_8$ singularity, each providing an $E_8$ type conformal matter. The presence of 12 hypermultiplets corresponds to the tuning of 8 out of the 20 hypers of $K3$ to realize the $E_8$ singularity.

This has also been described using an F-theory framework in [29, 54, 56]. Specifically, the $E_8 \times E_8$ heterotic string with instanton numbers $(12 + n, 12 - n)$ in $E_8 \times E_8$ has an F-theory realization on a CY 3-fold that is an elliptic fibration over $\mathbb{F}_n$ [57, 58]. The case of interest here corresponds to $n = 12$ and 24 point-like instantons, i.e., an elliptic fibration over $\mathbb{F}_{12}$. For the theory with the maximal number of tensor multiplets, the F-theory base can be obtained by performing 192 blowups. This involves fixing a fiber and blowing up its intersection with the $+12$ curve 24 times, thereby turning the $+12$ curve to $-12$ curve, which corresponds to making the $E_8$ instantons point-like. The resulting fiber corresponds to the configuration of a little string theory as $[\mathfrak{e}_8] \underbrace{1, 2, \cdots, 2, 1}_{25} [\mathfrak{e}_8]$ where the two $\mathfrak{e}_8$ are gauged by the two $-12$ curves in the global model. To obtain a Kodaira-type fiber, we need to perform two additional blowups (completing an LST). Thus, we first have a configuration in (4.2) with $N - 1 = 15$, and the additional blow-ups lead to (4.3) discussed above. As discussed there, the number of tensors generated by the blowups is $12N = 192$. Note that there is a canonical choice of $f, g$ from the toric action, and this choice gives Kodaira-type singularities. Hence this is a well-defined F-theory model. The construction of this class of toric bases was thoroughly examined in [56]. We may directly see that this geometric construction exactly saturates the tensor bound from the gravitational anomaly cancellation. Thus, the 6d supergravity theory that saturates the bound on the number of tensor multiplets is part of the string theory landscape as well.

## 4.2 Exact bound on Vectors

Let us now calculate the bound on the vector multiplets in 6d supergravity. We will put a bound on the rank of the vector multiplets $r(V)$. However, it is more convenient to first bound another number, which we denote as $\mathcal{V}$, given by the sum of the number of tensor multiplets and the rank of the gauge algebras, i.e., $\mathcal{V} \equiv T + r(V)$. In fact, this number, plus one for the Kaluza-Klein $U(1)$, gives the dimension of the Coulomb branch $\mathcal{K}_{5d}$ in the 5d supergravity under a circle compactification (without any twist), i.e., $\dim(\mathcal{K}_{5d}) = \mathcal{V} + 1$. For instance, the supergravity theory described in (4.4) has $\mathcal{V} = 489$.

The key idea for this calculation is once again to use the fact that all tensor multiplets supporting gauge algebras, except those for external generators $\mathcal{D}_i$ with $\mathcal{D}_i \cdot f > 0$ (which can support gauge algebras up to rank 20) and possibly those for $\mathcal{D}_i \cdot f = 0$ with $\mathfrak{g}_i = \mathfrak{g}_{\mathrm{small}}$ (with ranks no greater than 3), must be part of an LST in the charge class $f$ for an $H$-string. Therefore, the bound on $\mathcal{V}$ is mainly determined by the ranks of the gauge algebras, as well

as the number of tensor multiplets, in the LSTs with little string charge $f$. We will first examine the LST base structures that can yield a large $\mathcal{V}$, and demonstrate that LST bases that can potentially have a $\mathcal{V}$ larger than 489, the value for the theory in (4.4), are quite limited. Using optimization techniques, we will then show the bound

$$\mathcal{V} \leq 489 \ . \tag{4.6}$$

We first assert that LSTs with *frozen singularities* cannot be in the charge class $f$, and thus, we can exclude such LSTs from the discussions on LSTs for an *H*-string. All possible LSTs with frozen singularities are classified in [17] and can be constructed by compactifying F-theory on an elliptic Calabi-Yau 3-fold with frozen singularities. As previously noted, any LST for an *f*-string, which is the focus of our study, must have a '*P*'-type endpoint due to $f^2 = 0$ and $f \cdot b_0 = 2$. However, after investigating all allowed LSTs with frozen singularities classified in [17], we find that none of them has such an endpoint. The little strings in all these LSTs have charge $Q^2 = 0$ and $Q \cdot b_0 = 0$. For instance, an LST characterized by the gauge algebra $\mathfrak{so}_{20} \times \mathfrak{su}_{12} \times \mathfrak{sp}_2$, with three tensor multiplets (of which only two are dynamical) and bi-fundamental hypermultiplets for $\mathfrak{so}_{20} \times \mathfrak{su}_{12}$ and $\mathfrak{su}_{12} \times \mathfrak{sp}_2$, can be realized in a frozen phase of F-theory [17]. The charge class for a single little string is $Q = (1, 2, 2)$, which also represents the instanton numbers of the $\mathfrak{so}_{20}, \mathfrak{su}_{12}$, and $\mathfrak{sp}_2$ gauge algebras, respectively. This little string charge satisfies $b_0 \cdot Q = 0$, indicating that the endpoint of the LST is not of '*P*'-type.

Therefore, when discussing LSTs for an *H*-string below, we will focus exclusively on LSTs without frozen singularities, fully covered by the classification in [15], and ignore any cases involving frozen singularities. However, this does not mean that 6d supergravity cannot have matter fields associated with frozen singularities; it only means such fields cannot appear in an LST for the charge class $f$ where $f \cdot b_0 = 2$. In fact, some 6d supergravity theories arising from the frozen phase of F-theory were studied in [59], and indeed the LSTs in these examples are in the charge class $Q$ satisfying $Q^2 = 0$ and $Q \cdot b_0 = 0$.

We begin by analyzing LSTs with '*P*'-type endpoint that are constructed solely from E-type nodes connected by links. As computed in (A.5) and (A.6), where '*v*' refers to the $\mathcal{V}$ contribution from each link, it is clear that all interior links, including the half contributions from neighboring nodes, provide smaller $v$ compared to the $\mathfrak{e}_8 \otimes \mathfrak{e}_8$ conformal matter. It is important to note that the $\mathfrak{e}_8 \otimes \mathfrak{e}_8$ conformal matter has $\Delta(\mathfrak{e}_8 \otimes \mathfrak{e}_8) = v(\mathfrak{e}_8 \otimes \mathfrak{e}_8) = 30$. Therefore, as we embed $\mathfrak{e}_8 \otimes \mathfrak{e}_8$ conformal matter, the value of $v$ increases at the same rate as the gravitational anomaly contribution, i.e. $v/\Delta = 1$ for an $\mathfrak{e}_8 \otimes \mathfrak{e}_8$ conformal matter. It turns out that this is one of the most economical building blocks for an LST with a '*P*'-type endpoint. All other links connected to E-type nodes yield smaller $v$ contributions relative to their gravitational anomaly cost, i.e. $v/\Delta < 1$.

One might wonder if a side link could provide a larger $\mathcal{V}$ contribution, but this is not the case. The '$-1$' tensor multiplet in a side link attached to an E-type node cannot support a gauge algebra, and with this boundary condition, we find that no side link can achieve $v/\Delta = 1$. Additionally, when such a side link is added, the number of vector multiplets from external generators with $f \cdot \mathcal{D}_i > 0$ becomes smaller than that of $\mathfrak{e}_8 \times \mathfrak{e}_8$. We also note that any extra generator of type $\mathcal{C}_i \cdot f = 0$ supporting $\mathfrak{g}_i = \mathfrak{g}_{\text{small}}$, since their

contributions are bounded as $v/\Delta \leq 1/7$ (saturated by $\mathcal{C}_i^2 = -3$ with $\mathfrak{g}_i = \mathfrak{su}_3$), cannot provide additional room to increase the number of vector multiplets. Thus, supergravity theories embedding the LST of this type for an $H$-string or involving external generators associated with $\mathfrak{g}_i = \mathfrak{g}_{\text{small}}$ gauge algebras cannot reach the maximum value of $\mathcal{V}$.

In fact, the condition $v/\Delta \geq 1$, aside from a chain of $\mathfrak{e}_8 \times \mathfrak{e}_8$ conformal matters, can be achieved only when the LST base consists exclusively of '$-1$', '$-2$', and '$-4$' tensor multiplets. This is because only these tensor multiplets can accommodate $SU, Sp$ and $SO$-type gauge algebras with sufficiently large ranks. For example, an $\mathfrak{so}_{2n} \otimes \mathfrak{so}_{2n}$ conformal matter contributes $v = 2n - 4$ while $\Delta = 30$, yielding $v/\Delta \geq 1$ when $n \geq 17$. It is worth noting, however, that when D-type nodes are linked to E-type nodes, such large gauge algebras on D-type nodes cannot exist, as the boundary condition forces the '$-1$' tensor multiplets in an interior link connecting a D-type node to an E-type node to support either $\mathfrak{sp}_1$ or no gauge algebra, and the number of D-type nodes with other than $\mathfrak{so}_8$ gauge algebra is bounded by 5. Thus, only the LST bases consisting of '$-n$' tensor multiplets with $n = 1, 2, 4$ can support large rank $SU, Sp$ and $SO$-type gauge algebras.

All possible SCFTs and LSTs with only $SU, Sp$ and $SO$-type gauge algebras, which are called "semi-classical" configurations, are classified in [13, 14]. Among these, 9 LST configurations have '$P$'-type endpoints, with 5 configurations having a short base as

$$
\begin{aligned}
&\textbf{\textit{1}}) \quad 4\ 1\ \overset{1}{4}\ 1\ 4\ 1\ 4\ 1\ , \qquad \textbf{\textit{2}}) \quad 1\ 4\ 1\ \overset{1}{4}\ 1\ 4\ 1\ , \\
&\textbf{\textit{3}}) \quad 2\ 2\ 1\ 4\ 1\ , \qquad \textbf{\textit{4}}) \quad 2\ 2\ 2\ 1\ 4\ , \qquad \textbf{\textit{5}}) \quad 2\ 1\ 2\ ,
\end{aligned}
\tag{4.7}
$$

and 4 configurations have a long base as

$$
\begin{aligned}
&\textbf{\textit{6}}) \quad 1\ \overset{1}{4}\ 1\ 4\ 1\ 4\ 1\ \cdots\ 4\ 1\ \overset{1}{4}\ 1\ , \qquad \textbf{\textit{7}}) \quad 1\ \overset{1}{4}\ 1\ 4\ 1\ 4\ 1\ \cdots\ 4\ 1\ 2\ , \\
&\textbf{\textit{8}}) \quad 2\ \overset{2}{2}\ 2\ 2\ \cdots\ 2\ 1\ , \qquad \textbf{\textit{9}}) \quad 1\ 2\ 2\ 2\ \cdots\ 2\ 1\ ,
\end{aligned}
\tag{4.8}
$$

where $\cdots$ represents a repeating pattern of tensor multiplets. Here, each '$-1$', '$-2$', '$-4$' tensor multiplet supports $\mathfrak{sp}$, $\mathfrak{su}$, and $\mathfrak{so}$ gauge algebras, respectively. There is also a bi-fundamental hypermultiplet between intersecting '$-1$' and '$-2$' tensor multiplets, and a bi-fundamental half-hypermultiplet between intersecting '$-1$' and '$-4$' tensor multiplets.

We investigate each of these 9 LST configurations using a numerical optimization. For this, we assume that all flavor symmetries in the LSTs are gauged by gauge algebras with the largest permissible dimension, possibly including additional tensor multiplets. This setup provides the maximum number of vector multiplets, which contribute more negatively to the gravitational anomaly. This in turn allows the LSTs to support more tensor multiplets with larger gauge algebras. We then explicitly construct supergravity theories embedding these LSTs to check if they can yield the maximum $\mathcal{V}$. The solutions for such LSTs from the numerical optimization are summarized in Appendix B.

As a result, we find that the following tensor configuration for a supergravity theory

achieves the maximum $\mathcal{V}$:

$$
\begin{array}{c}
\mathfrak{sp}_{40} \\
1 \\
\begin{array}{ccccccccc}
\mathfrak{so}_{64} & \mathfrak{sp}_{56} & \mathfrak{so}_{176} & \mathfrak{sp}_{72} & \mathfrak{so}_{128} & \mathfrak{sp}_{48} & \mathfrak{so}_{80} & \mathfrak{sp}_{24} & \mathfrak{so}_{32} \\
4 & 1 & 4 & 1 & 4 & 1 & 4 & 1 & (4)
\end{array}
\end{array} \ . \tag{4.9}
$$

In this theory, an LST of the first type in (4.7) is embedded, along with an additional '$-4$' tensor multiplet, denoted as (4), which is connected to the last tensor multiplet in the LST through a bi-fundamental half-hypermultiplet for $\mathfrak{so}_{24} \times \mathfrak{so}_{32}$. This supergravity theory contains a total rank of 480 in gauge algebras and 9 tensor multiplets, resulting in $\mathcal{V} = 489$. One can readily observe that any attempt to add external gauge algebras or matter fields to this theory would lead to a decrease in $\mathcal{V}$. This is the maximum $\mathcal{V}$ for supergravity theories containing LSTs in semi-classical configurations with large $SU$, $Sp$, and $SO$-type gauge algebras.

Quite remarkably, this maximal theory also has a string theory realization, discussed in [29], where it describes 24 point-like instantons in the $SO(32)/\mathbb{Z}_2$ heterotic string theory on $K3$ with an $E_8$ singularity, including 12 neutral hypermultiplets. Note that this base may be obtained from a Hirzebruch surface $\mathbb{F}_4$ by performing eight point blowups, with specific choices of $f$ and $g$ to give the required gauge group. While this example has the same Hodge numbers for the corresponding elliptic CY as the one constructed from the $E_8 \times E_8$ string discussed in the previous subsection, their bases are different. Indeed, the two LST's are T-dual of one another [60, 61] under a circle reduction. These two theories represent different theories in 6d but yield the same 5d reduction.

Combining this with the $\mathcal{V}$ bound for theories containing LSTs on a long base involving exceptional gauge algebras, which is saturated by the theory in (4.4), we find that the maximum $\mathcal{V}$ is 489, as stated in (4.6). This, along with the optimization results in Appendix B, also leads to the conclusion that the maximum possible rank of gauge algebras in 6d supergravity is 480, i.e.,

$$
r(V) \le 480 \ . \tag{4.10}
$$

The underlying reason for this is that the maximum $r(V)$ cannot be reached by theories with $T > 9$ due to the $\mathcal{V}$ bound, and a higher rank of gauge algebras, when $T \le 9$, is possible only if the theories contain semi-classical LSTs solely with $SU$, $Sp$, and $SO$-type gauge algebras. Then, our numerical optimization for such theories reveals that the supergravity theory with the highest possible rank of gauge algebras is exactly the one given in (4.9), with $r(V) = 480$.

Thus, we identify two supergravity theories that saturate the $\mathcal{V}$ bound in (4.6): one in (4.4) and another in (4.9). Additionally, the theory in (4.9) also saturates the rank bound in (4.10). Both of these theories are realized in string theory, providing significant evidence supporting the string lamppost principle.

### 4.2.1 $T = 9$ with $b_0 = 0$ theories

These theories have an even charge lattice whose BPS cone is only generated by two types of generators: one with $\mathcal{C}_i^2 = -2$ and $k_L = 0$, and another one with $\mathcal{C}_i^2 = 0$ and $k_L = 1$.

Particularly, the latter corresponds to the charge classes associated with Type II strings with central charges $c_L = c_R = 4$. We will demonstrate that such theories always include the charge classes for Type II strings in their spectra. Using these strings, we will then determine the bound on the rank of the gauge algebras.

To begin, we observe that theories involving only these two types of generators include a sufficient number of charged (fundamental or bi-fundamental) hypermultiplets to fully Higgs the gauge algebras. Even gauge algebras only with half-hypermultiplets, such as $\mathfrak{so}_7$ and $\mathfrak{g}_2$, can be Higgsed. After performing the Higgsings, we can have '$-2$' generators (or M-strings) without associated gauge algebra. However, as previously discussed, since these generators exhibit a Weyl reflection as they shrink, the tensor branch can be extended to remove all such '$-2$' generators. The remaining generators then satisfy $\mathcal{C}_i^2 = 0$ and $k_L = 1$, which correspond to Type II strings. Thus, all supergravity theories of this type contain the charge classes for Type II strings both before and after the Higgsings. In fact, in this case, the BPS cone coincides with the tensor cone, as all generators satisfy $\mathcal{C}_i^2 = 0$, and we have infinitely many such generators that intersect with all other generators. This also means that every null generator can shrink at an infinite distance in the moduli space.

We can assign gauge algebras to these null generators. However, because each null generator must intersect with another (distinct) null generator that has a central charge $c_L = 4$, the rank of the gauge algebra on any single generator cannot exceed 4, as discussed in (2.14). An additional restriction on the rank arises from the gravitational anomaly cancellation condition. Given that the number of tensor multiplets is fixed at 9, the difference between the number of charged hypermultiplets and vector multiplets cannot be bigger than $273 - 29 \times 9 = 12$. From the matter content listed in Table 4, we observe that each gauge algebra on a null generator with $k_L = 1$ contributes $H - V = 1$. This fact and the gravitational anomaly constraint imply that we can have at most 12 independent gauge algebras. However, if a null generator hosts a gauge algebra, another null generator cannot, because they always intersect, but no charged hypermultiplet exists that carries representations for two distinct gauge algebras in this case. As a result, only one gauge algebra can be supported on a single null generator, and thus the total rank of the gauge algebras in this case is bounded by 4.

We now introduce gauge algebras over '$-2$' generators. Choosing a null generator $S$, among those identified above, the '$-2$' generator $\mathcal{C}_i$ with a gauge algebra can satisfy either $\mathcal{C}_i \cdot S > 0$ or $\mathcal{C}_i \cdot S = 0$. In the first case, the rank of the gauge algebra on $\mathcal{C}_i$ is again bounded by 4 due to the current algebra constraint from the Type II string associated with $S$. In the second case, using the same argument as for an $H$-string in Section 3.3, we claim that $\mathcal{C}_i$ is a BPS generator for an LST in the charge class $S$, or otherwise it supports a gauge algebra $\mathfrak{g}_i = \mathfrak{g}_{small}$. The LSTs, as classified in [14, 15], formed only by '$-2$' tensor multiplets contain $SU$-type gauge algebras that are arranged in an affine ADE-type Dynkin diagram. The vectors and charged hypermultiplets in each such LST contribute $N + 1$ to the gravitational anomaly for LST bases in affine $SU(N + 1)$, $SO(2N)$, and $E_N$ configurations. Moreover, each LST of this kind must intersect other null charges as two distinct null charges always intersect. This implies that at least one '$-2$' tensor multiplet in the affine type base must intersect with another null charge, among those we identified

– 32 –

after the Higgsings, and thus the rank of its gauge algebra is bounded by 4. Once the rank of an $\mathfrak{su}$ gauge algebra in such an LST is fixed, all other gauge algebras in the LST are automatically determined. Consequently, the total rank of any LST embedded in this type of supergravity is constrained.

We find that the configuration embedding only a single LST with an affine $E_8$ type base, consisting of nine '$-2$' tensor multiplets, as given by

$$\overset{\displaystyle \overset{\mathfrak{su}_{3N}}{2}}{\underset{\underset{2}{\mathfrak{su}_N}\ \underset{2}{\mathfrak{su}_{2N}}\ \underset{2}{\mathfrak{su}_{3N}}\ \underset{2}{\mathfrak{su}_{4N}}\ \underset{2}{\mathfrak{su}_{5N}}\ \underset{2}{\mathfrak{su}_{6N}}\ \underset{2}{\mathfrak{su}_{4N}}\ \underset{2}{\mathfrak{su}_{2N}}}{}} \,, \tag{4.11}$$

can support the maximal rank of gauge algebras. In this case, because the LST lacks any flavor symmetry, none of the null generators intersecting this LST can carry a gauge algebra unless $N = 1$. As discussed, the rank of at least one gauge algebra is bounded by 4. This means the maximal rank in this configuration is achieved by setting the first gauge algebra to this upper limit, specifically $\mathfrak{su}_5$ for the first tensor multiplet. Thus, the highest achievable rank for the gauge algebras in this LST is 141. The supergravity theory embedding a single LST of this type together with one additional '$-2$' or null generator without gauge algebra, though it is unclear whether a consistent supergravity theory with this configuration can be realized, is the maximal theory for $T = 9$ and $b_0 = 0$, with $V = 2991, H_0 = 3$ and $r(V) = 141$.

In summary, the rank of gauge algebras in supergravity theories with $T = 9$ and $b_0 = 0$ is bounded by 141.

### 4.2.2 $T = 0$ theories

For theories without tensor multiplets, our method for constraining massless fields is not applicable. Instead, we can utilize the unitarity condition for supergravity strings, as given in (2.14), to impose a bound on the rank of gauge algebras. We will find that this gives an upper bound of $r(V) = 32$.

For $T = 0$, the BPS cone is one-dimensional and generated by a single vector, which we denote as $\mathcal{C}$. For a one-dimensional unimodular lattice, we can take the charge vectors to be $\mathcal{C} = 1$ and $b_0 = 3$, where $b_0 \cdot \mathcal{C} = 3$. According to the attractor mechanism, for any sufficiently large multiple of the generator, say $m\mathcal{C}$ with $m \gg 1$, there exists a corresponding supersymmetric black string solution. As a result, large charge states are filled by BPS strings. However, to determine an exact bound on the rank of gauge algebras, we need to know the smallest possible $m$ for supergravity strings, which is challenging from the EFT framework.

We now argue that, for every supergravity theory with $T = 0$, the smallest charge $\mathcal{C}$, specifically $m = 1$, is realized by a BPS string. The reasoning for this claim is based on the idea from [62], which involves 5d compactifications and an in-depth analysis of the 5d Coulomb branch moduli space, as explained below. First, consider a compactification on a circle (without twist) to 5d, while preserving 8 supersymmetries. This compactification leads to a 5-dimensional theory with a Coulomb branch moduli space of dimension $r(V) + 1$, where $r(V)$ is the rank of the 6d gauge algebra. This moduli space is characterized by a

so-called prepotential, expressed as a cubic function:

$$\mathcal{F} = \frac{1}{6} C_{IJK} t_I t_J t_K \ , \tag{4.12}$$

with the constraint $\mathcal{F} = 1$, where $C_{IJK} \in \mathbb{Z}$ represents the cubic Chern-Simons coefficients, and $t_I$ for $I = 0, 1, \ldots, r(V) + 1$ are the expectation values of the scalar fields in the 5d vector multiplets. Importantly, the metric on the Coulomb branch is determined by taking the second derivative of the prepotential, such that $M_{IJ} = \partial_{t_I} \mathcal{F} \partial_{t_J} \mathcal{F} - \partial_{t_I} \partial_{t_J} \mathcal{F}$. The prepotential for 5d theories arising from compactifying 6d theories on a circle has been derived in [63, 64]. Therefore, the metric of the moduli space in the 6-dimensional theory on a circle can be computed in a systematic manner.

A distinctive feature of 5d theories originating from 6d supergravity theories with $T = 0$, regardless of gauge algebra type, is that BPS particles stemming from the ground states of 6d BPS strings can become massless at a finite-distance boundary on the Coulomb branch moduli space, and that when it happens, the moduli space metric degenerates and has a vanishing eigenvalue, as investigated in [62]. This vanishing eigenvalue, located at a finite distance in the moduli space along a one-dimensional direction, implies that the coupling for the $U(1)$ gauge charge in this direction diverges and it thus implies an emergent 5d rank-1 SCFT at that location, as discussed in works like [65, 66]. This provides a natural 5-dimensional generalization of our assumption that each finite-distance boundary of the moduli space hosts a strongly coupled SCFT, though a flop transition could may occur instead in 5d theories.

The $U(1)$ gauge charge associated with the zero eigenvalue is specifically given by $q \equiv q_0 - 3q_1$, where $q_0$ denotes the KK-charge and $q_1$ represents the string charge. We can express the prepotential in terms of the parameter $t_c$ for this charge $q$ as shown in [62]:

$$6\mathcal{F} = 9t_c^3 + \bar{\mathcal{F}}(t_I) \ , \tag{4.13}$$

where $\bar{\mathcal{F}}(t_I)$ is independent of the SCFT parameter $t_c$ and remains finite at the boundary as $t_c \to 0$. The coefficient '9' for $t_c^3$, which represents the cubic Chern-Simons term of the local SCFT, allows us to identify the precise SCFT at the boundary $t_c \to 0$. Assuming the 5d rank-1 SCFT classification provided in [67–69] is complete, similar to our assumption for 6d SCFTs and LSTs, the local SCFT here is the $E_0$ theory. This result shows that every 6d supergravity theory with $T = 0$ includes an $E_0$ theory within its Coulomb branch moduli space after a circle compactification without twist.

The BPS spectrum of the 5d $E_0$ SCFT, as computed, for example, in [70, 71], reveals that a BPS state with the minimal charge $q = -3$ exists. This means that, in the 6d theory before compactification, the ground state of a BPS string with string charge $q_1 = 1$ must also exist. Therefore, we assert that the minimum string charge $q_1 = 1$ in the charge lattice of any 6d supergravity theory with $T = 0$ is indeed occupied by a BPS string.

This minimal BPS string enables us to determine a bound on the rank of gauge algebras. As this string corresponds to a supergravity theory with central charge $c_L = 32$, we conclude that the highest rank for gauge algebras, including $U(1)$ factors, in 6d theories without tensor multiplets is 32. Also, in the absence of $U(1), SU(2)$, and $SU(3)$ gauge symmetries, a stricter bound of 24 applies, as shown in [10].

## 4.3 Exact bound on Neutral hypermultiplets

Once the massless matter content charged under gauge algebras is fixed, the number of neutral hypermultiplets, $H_0$, is determined by the gravitational anomaly formula

$$H_0 = 273 - H_c + V - 29T = 273 - \Delta , \qquad (4.14)$$

where $H_c$ represents the number of charged hypermultiplets, and $\Delta \equiv H_c - V + 29T$ collectively denotes the gravitational anomaly contribution excluding $H_0$. Hence, the upper bound on $H_0$ is set by the minimum $\Delta$.

In order to find the minimum $\Delta$ for 6d supergravity, we first compute the minimum $\Delta$ for 6d LSTs. For this purpose, we will employ "*Higgsing*" of LSTs. Here, the term *Higgsing* involves both the Higgsing of gauge symmetries and also small instanton transition from a tensor multiplet to 29 hypermultiplets. As shown in [13], any 6d SCFT without frozen singularities can always be Higgsed down to a minimal SCFT corresponding to its endpoint consisting of non-Higgsable clusters. Likewise, we can prove that any LST without frozen singularities can be Higgsed to its endpoint. The proof closely parallels that of the SCFT cases, and the key point is that LST bases, like SCFT bases, are non-compact and have an infinite dimensional space of holomorphic functions $f, g$ to tune for the Weierstrass model. A generic choice would make all the elliptic fibers over '$-1$' curves nonsingular, and we may shrink these curves and further Higgs one by one and arrive at the endpoint for each LST. For an $H$-string in the charge class $f$, the endpoint is a trivial LST with no tensor multiplet or gauge algebra, corresponding to a critical heterotic string.

We remark that the Higgsing of LSTs cannot increase $\Delta$, and thus $\Delta$ for the trivial LST is the minimum, i.e., $\min(\Delta_{\mathrm{LST}}) = 0$. This is evident for any Higgsing of gauge algebra, as gauge fields get mass and Higgsed by eating up an equal number of charged hypermultiplets, resulting in $\Delta_{\mathrm{LST}}^{\mathrm{before}} \geq \Delta_{\mathrm{LST}}^{\mathrm{after}}$. Also, a small instanton transition, when available, trades a tensor multiplet into 29 hypermultiplets, which cannot increase $\Delta$. This shows that $\Delta$ for LSTs related by Higgsings can only decrease under the RG-flows.

When we embed LSTs into supergravity, the Higgsing argument for SCFTs and LSTs may no longer hold. Additional supergravity matter fields and gaugings could prohibit such Higgsing options. So we do not know if it is possible all LSTs in the charge class $f$ can be Higgsed to a trivial LST in a supergravity. However, we can still use the fact that the minimum $\Delta_{\mathrm{LST}}$ from any LST sector embedded in the supergravity is $\min(\Delta_{\mathrm{LST}}) = 0$.

We now need to account for the gravitational anomaly contributions $\Delta_{\mathrm{ext}}$ from external gauge algebras on charges $b_i$. As previously discussed, the maximum rank of these gauge algebras on charges with $f \cdot b_i > 0$ is 20. This implies that the minimum possible $\Delta_{\mathrm{ext}}$ would be $-820 + 29 = -791$, from an $\mathfrak{sp}_{20}$ gauge algebra on a tensor multiplet with charge $Q$ satisfying $f \cdot Q > 0$. Note that the external gauge algebras of type $\mathfrak{g}_{\mathrm{small}}$ on $f \cdot b_i = 0$ can only increase $\Delta_{\mathrm{ext}}$, and thus we can ignore them. Other gauge algebras, such as $\mathfrak{so}_{40}$ or $\mathfrak{e}_8 \times \mathfrak{e}_8 \times \mathfrak{f}_4$, yield larger values of $\Delta_{\mathrm{ext}}$ than $\min(\Delta_{\mathrm{ext}}) = -791$. Here, we have omitted the contributions from hypermultiplets charged under these external gauge algebras, as these would only increase $\Delta_{\mathrm{ext}}$ and may already be counted in the LST contributions. The gravitational anomaly contribution from an $\mathfrak{sp}_{20}$ gauge algebra is likely bigger than $-791$

due to its coupling with charged hypermultiplets that we have omitted, so we do not claim that an external $\mathfrak{sp}_{20}$ gauge algebra minimizes $\Delta_{\mathrm{ext}}$; rather, we assert that while other external gauge algebra and tensor multiplet configurations might minimize $\Delta_{\mathrm{ext}}$, none of them can yield a value smaller than $\min(\Delta_{\mathrm{ext}}) = -791$. Therefore, the number of neutral hypermultiplets in 6d supergravity is bounded as

$$H_0 \leq 273 - \min(\Delta_{\mathrm{LST}}) - \min(\Delta_{\mathrm{ext}}) = 273 + 0 + 791 = 1064 . \tag{4.15}$$

For $T = 0$ theories, the maximum $H_0$ is 273 and smaller than this bound. This is due to the fact that, to satisfy gauge anomaly cancellation conditions, any gauge algebra in these theories, which are supported on charges $Q$ with $Q^2 > 0$, requires more charged hypermultiplets than vector multiplets, and thus the configuration without gauge fields yields the maximum number of neutral hypermultiplets, which is 273.

The above bound can be improved if LSTs embedded in supergravity can be Higgsed down to their endpoints. In such cases, the LSTs in the charge class $f$ we previously identified can be Higgsed to trivial LSTs, even after coupling to gravity. This yields a $T = 1$ theory with an $H$-string that carries no gauge algebra, which has $\Delta_{\mathrm{LST}} = 0$. The maximum external gauge symmetry in this $T = 1$ theory is an $E_8$ symmetry on a '$-12$' tensor multiplet intersecting the $H$-string, which provides $\Delta_{\mathrm{ext}} = -248$. This theory then has $273 + 248 - 29 = 492$ neutral hypermultiplets. As this theory is produced via Higgsings, and Higgsings (including small instanton transitions) can only increase the number of neutral hypermultiplets, as discussed earlier, no other theory before the Higgsing can have more neutral hypermultiplets. Thus, we derive the bound

$$H_0' \leq 492 . \tag{4.16}$$

Here, $H_0'$ now represents the bound when all LSTs are Higgsable to their endpoints, even after coupling with supergravity. All F-theory models exhibit this property, which follows from the fact that on a Hirzebruch surface (including $\mathbb{F}_0 = \mathbb{P}^1 \times \mathbb{P}^1$), a generic point of hypermultiplet moduli Higgs the little string wrapping the fiber to the endpoint. Thus, this bound holds for all geometric models.[14]

## 4.4 Swampland Examples

The structure we have uncovered for consistent 6d supergravity theories allows us to set bounds on the number of massless fields, and thereby exclude an infinite number of effective field theories that might otherwise appear consistent. In this sense, this structure serves as a new form of quantum anomaly. We can use it to differentiate low-energy theories that can arise as the IR limit of a quantum gravitational theory from those that are ultimately inconsistent with gravity and thus fall into the Swampland. Indeed, this structure imposes significant constraints on general 6d supergravity theories, even within the bounds on massless fields. In this subsection, we will illustrate the extensive implications of this

---

[14]We can also see this directly from geometry: any coefficients $f, g$ in the Weierstrass model survive under blowing down, hence blowing down never decreases the number of hypermultiplets.

| $G_i$ | $H_i$ | $b_i^2$ | $b_0 \cdot b_i$ |
|:---:|:---:|:---:|:---:|
| $\mathfrak{su}_8$ | $16 \times \mathbf{8} \oplus 3 \times \mathbf{28}$ | $1$ | $3$ |
| $\mathfrak{su}_{16}$ | $8 \times \mathbf{16} \oplus \mathbf{136}$ | $-1$ | $-1$ |
| $\mathfrak{sp}_4$ | $16 \times \mathbf{8}$ | $-1$ | $1$ |

**Table 3**. Gauge algebras for a class of anomaly-free supergravity theories.

structure in the context of the Swampland program through concrete anomaly-free examples that strictly satisfy all the bounds but still turn out to be inconsistent.

We start by arguing that any supergravity theory with $T > 0$ and gauge algebras $G_i$ supported on time-like charges $Q_i$ satisfying $Q_i^2 > 0$ is inconsistent if the total rank of $G_i$'s is bigger than 20. This is due to the fact that an $H$-string intersects all time-like charges, and the unitarity constraint (2.14) applied to the $H$-string enforces an upper bound of 20 on the total rank of the gauge algebras associated with these $Q_i$'s. On the other hand, as we discussed in the previous section, the rank bound is 32 when $T = 0$.

Also, the same bound applies to the $\mathfrak{su}_N$ gauge algebra for $N \geq 8$ on a charge $Q$ with $Q^2 = -1$ and $b_0 \cdot Q = -1$, realized through a frozen singularity. We note that every $H$-string must intersect with such a charge $Q$, as otherwise this $Q$ would belong to an LST associated with the $H$-string, which is not allowed. According to the classification of LSTs [15, 17], such charges $Q$ can only appear in LSTs with non-'$P$'-types, whereas the $H$-string endpoint is always '$P$'-type. Thus, we find the bound $N \leq 21$ in this case as well.

For instance, a class of anomaly-free supergravity theories including $\mathfrak{su}_8, \mathfrak{su}_{16}$, and $\mathfrak{sp}_4$ gauge algebras on $b_i^2 = 1, -1, -1$ and $b_0 \cdot b_i = 3, -1, 1$, respectively, is proposed in [10]. The anomaly vectors and the charged matter content are summarized in Table 3. These gauge algebras can be interconnected by carefully arranging the charged hypermultiplets and their mutual intersections $b_i \cdot b_j$ for $i \neq j$. For example, three $\mathfrak{su}_8$ gauge algebras can be connected via bi-fundamental hypermultiplet for each pair, with intersections $b_1 \cdot b_2 = b_2 \cdot b_3 = b_3 \cdot b_1 = 1$. However, since this arrangement places gauge algebras on $b_i^2 = 1$ with a total rank of 21, the theory is inconsistent unless $T = 0$. Likewise, all such theories containing multiple $\mathfrak{su}_8$ and $\mathfrak{su}_{16}$ gauge factors with a total rank bigger than 20 fall into the Swampland.

Another interesting examples are supergravity theories with gauge algebra $G = (\mathfrak{e}_8)^n$. An infinite class of anomaly-free theories of this type, characterized by particular tensor intersection forms and anomaly vectors, was proposed in [31], but these were later excluded in [8] by applying constraints from the unitarity of BPS strings. However, it remains plausible that other configurations of intersection forms and anomaly vectors could satisfy the known unitarity conditions. Here, we apply our constraints to examine such theories and show that most of them fail to be consistent.

An $\mathfrak{e}_8$ gauge algebra comes with a '$-12$' tensor multiplet due to the anomaly cancellation condition, which requires the theory to have at least $n$ tensor multiplets. This guarantees the presence of an $H$-string, say $f$, in the spectrum. Each tensor multiplet can either intersect $f$ or not intersect at all. In the former case, since the rank of gauge algebras from tensor multiplets intersecting $f$ is bounded by 20, we can have at most two such $\mathfrak{e}_8$ gauge algebras. The remaining $\mathfrak{e}_8$ gauge algebras reside on tensor multiplets that do not intersect

$f$, and thus must be elements of LSTs in the charge class $f$. A single LST without other types of gauge algebras can accommodate only one $\mathfrak{e}_8$ gauge algebra, and this is possible only if the LST corresponds to an $\mathfrak{e}_8$ theory with 12 small instantons forming its base as

$$2\ 2\ 2\ 2\ 2\ 2\ 2\ 2\ 2\ 2\ 2\ 1\ \overset{\mathfrak{e}_8}{\underline{12}}\ . \tag{4.17}$$

Note that this LST cannot intersect with another '$-12$' tensor multiplet for an $\mathfrak{e}_8$ gauge algebra, as an M-string for '$-2$' tensor multiplets can only intersect with a tensor associated with a rank-1 gauge algebra, and the E-string for the '$-1$' tensor already intersects with an $\mathfrak{e}_8$ tensor multiplet. Therefore, when the theory contains an LST of this type in the charge class $f$, no '$-12$' tensor multiplet can positively intersect with $f$. Moreover each LST of this type contributes $12 \times 29 - 248 = 100$ to the gravitational anomaly. Thus, we can have at most two such LSTs in a supergravity theory. This proves that the supergravity theories with gauge algebra $G = (\mathfrak{e}_8)^n$ can accommodate at most two $\mathfrak{e}_8$ gauge algebras, i.e., $n \le 2$, and thus the number of tensor multiplet is bounded by $T \le 26$, regardless of the specific choices of tensor intersection forms or anomaly vectors. Indeed, the theory at $T = 25$ with $\mathfrak{e}_8 \times \mathfrak{e}_8$ gauge algebra is realized via the compactification of M- theory on K3$\times(S^1/\mathbb{Z}_2)$ with 24 M5-branes along the interval, as shown in [72]. Interestingly, however, the bound $n \le 2$ allows up to 26 tensor multiplets. This opens up a fascinating direction as a small Swampland program: determining whether the theory with $T = 26$ and $\mathfrak{e}_8 \times \mathfrak{e}_8$ gauge algebra can exist.

## 5 Geometric Interpretation

The arguments in the previous sections did not assume any specific realization of $\mathcal{N} = (1,0)$ supergravity theories, and thus provided a bottom-up derivation of the constraints. In this Section, we review what is known from stringy constructions and compare our derivations with the geometric features of string constructions.

The most prominent construction of these models involves F-theory on elliptic CY 3-folds [57, 58]. In this context, as we will review, the two assumptions we made are true geometrical facts and the physical derivation in the previous sections are easy to interpret mathematically and are essentially rigorous, except for the constraints from the anomaly cancellation condition. Thus, our bounds can be viewed as geometrical bounds on elliptic Calabi-Yau manifolds, which were not known before. Even though, as we will review, the finiteness of elliptic CY manifolds is known, the bounds on the Hodge numbers $h^{1,1}(\text{CY}) \le 491$, $h^{1,1}(\text{Base}) \le 194$ we derive for elliptic 3-folds using physical reasoning are new. The other bound $h^{2,1}(\text{CY}) \le 491$, on the other hand, was known by Taylor in [73], and implies the $h^{1,1}(\text{CY})$ bound when mirror symmetry is applicable, as commented in [74]. Moreover, as we have already mentioned, the bounds are sharp, with examples of Calabi-Yau manifolds that achieve it.

Consider F-theory compactification on an elliptic Calabi-Yau 3-fold $\pi : X \longrightarrow B$ where $B$ is the bases of the 3-fold, and the 'visible' part of the geometry from the Type IIB

perspective. We will also assume that we have a section for this fibration.[15] The BPS strings in 6d come from D3-branes wrapping on holomorphic 2-cycles, hence the BPS cone in the 6d supergravity is the effective cone on $B$. Tensor moduli space comes from a Kähler structure on $B$, hence the tensor cone is the Kähler cone of $B$ and its closure including SCFT points and infinite distance points is mathematically known as the nef[16] cone of $B$. The anomaly coefficients are given by $b_0 = c_1(B)$ and $b_i$ are represented by holomorphic divisors $D_i$ in the base. The physical assumption that tensionless BPS strings (corresponding to CFT, LST or critical strings) emerge at any boundary of the 6d tensor moduli space follows from the duality between effective cone and nef cone (Kleiman's criterion).

Kodaira canonical bundle formula of elliptic fibration says $0 = K_X = \pi^*(K_B + \sum \mu_i D_i + M)$ where $\pi$ is the projection from $X$ to $B$, $D_i$ are the discriminant loci (7-brane loci) on $B$ and $\mu_i$ are the log canonical threshold (deficit angle around $D_i$). $M$ is the moduli part of this fibration which characterizes the curvature on the base $B$, satisfying $12M = j^*\mathcal{O}_{\mathbb{P}^1}(1)$, where $j : B \longrightarrow \mathbb{P}^1$ is the $j$-invariant, hence it is non-negative (more precisely nef) and vanish if and only if the elliptic fibration is locally constant. This corresponds to the Kodaira condition in F-theory literature. This can be viewed as the condition of Ricci-flatness, as studied for example in [57, 58, 75].

As we have seen from the physics discussion, there are two essentially different cases to consider: $c_1 \neq 0$ and $c_1 = 0$.[17]

Complex surfaces with vanishing $c_1$ have three possibilities: complex torus, K3 and Enriques, which preserve 32, 16, 8 supercharges respectively. So for 6d $\mathcal{N} = (1,0)$ supergravity or strictly the one with lower supersymmetry, we have irreducible elliptic Calabi-Yau threefold corresponding to an Enriques surface. Moreover, the canonical bundle formula implies that there are no 7-branes and that the axion-dilaton is constant, i.e., this must be a free action orientifold of Type IIB string theory on a $K3$, with $H = 12$, $V = 0$ and $T = 9$. We may directly check that the anomaly cancellation is satisfied. The BPS/nef cone in this case is generated by the positive roots of the hyperbolic $E_{10}$ root system, which is a fundamental domain of the Weyl group $W(E_{10})$ acting on the future cone. Any subset of $E_{10}$ Dynkin diagram gives a boundary strata with (2,0) SCFT/LST degree of freedom, determined by this subset viewed as a Dynkin diagram.

So the essential problem is to understand the $c_1 \neq 0$ case. The pair $(B, D = \sum \mu_i D_i)$ satisfy $M = c_1 - D$ is nef and non-zero. Such a structure is called a semi-Fano pair. The metric on $B$ satisfies that the Ricci curvature is non-negative everywhere and has conical singularity at $D_i$ with deficit angle $2\pi\mu_i$. Alexeev [76] proved such $(B, D)$ are bounded, as a natural generalization of the boundedness of Fano manifolds. Here, being bounded means not only the topological types are finite but also the complex structures are parameterized by a finite type moduli space, i.e., by finitely many families. This produces a bounded family of Weierstrass models over these bases, hence a bounded family of elliptic Calabi-Yau threefold

---

[15]Having a section or not becomes more physical in the 5d compactifications, and will not be physically relevant for 6d, and so we assume the simpler setup where we have a section.

[16]A nef divisor is a divisor which intersects non-negatively with every effective curve.

[17]Here we neglect the torsion part of $H^2(B)$ which is not relevant (and can be viewed as a discrete gauge symmetry) and view $c_1 \in H_2(B)^\vee \simeq H^2(B)/\text{Torsion}(H^2(B))$.

with a rational section (up to flops). Moreover, Gross [77] showed the boundedness without assuming the existence of a rational section by analyzing the Tate-Shafarevich group, which corresponds to the discrete gauge group in the 6d F-theory. Birkar, Cerbo, and Svald [78] proved the finiteness of irreducible elliptic Calabi-Yau $n$-fold in any dimension by extending the boundedness of semi-Fano pair, based on Birkar's work [79] on the boundedness of Fano pairs and Fano-type fibrations. This in particular implies that the $\mathcal{N} = 1$, $d = 4$ string landscape coming from F-theory on elliptic 4-folds is also finite.

To prove such boundedness results, various techniques are applied: Mori et.al. [80, 81] developed a famous 'bend and break' method to produce families of rational curves on a Fano manifold $X$ with bounded degree, which gives universal bound on $c_1(X)^n$ for the Fano manifold, and Kodaira embedding via $K_X^{-1}$ realize $X$ as a submanifold of $\mathbb{P}^N$ for some fixed $N$, which forms a bounded family. But this 'bend and break' method does not apply to semi-Fano pair $(B, D)$. Generalizing to semi-Fano pairs requires sophisticated induction on dimension and the full machinery of birational geometry.

Alexeev [76] used the diagram method developed by Nikulin [82] to prove the boundedness of $(B, D)$ for the case where $B$ is two-dimensional. The basic idea is to study the combinatorics of diagrams spanned by exceptional curves (similar to what we did for SCFT and LST), but a special role is played in addition by the minimal hyperbolic diagrams called Lanner diagrams, which are minimal curve configurations that are not shrinkable at finite or infinite distance. These diagrams have finite combinatorial types up to blow-up, and after we impose the Kodaira condition finitely many blow-ups are allowed. So the possible minimal non-shrinkable configuration of exceptional curves (i.e., Lanner diagrams of exceptional curves) are finite. Then the combinatorics of polyhedron in a hyperbolic space guarantee that the rank of the Picard group as well as $(c_1 - D)^2$ are bounded, and the same argument using Kodaira embedding into $\mathbb{P}^N$ shows that such $(B, D)$ are bounded.

Our physical approach is essentially different from these mathematical ones. Recall our three main claims:

---

**Claims**

1. Every 6d (1,0) supergravity with $T \geq 1$ (except for a special isolated case with $T = 9$ and $b_0 = 0$) must contain an *H-string* with charge $f$,

$$f^2 = 0 \ , \quad b_0 \cdot f = 2 \ , \quad k_L(f) = 0 \ , \quad f \cdot \mathcal{C}_i \geq 0 \tag{5.1}$$

   for all generators $\mathcal{C}_i$ of the BPS cone.

2. Any generator $\mathcal{C}_i$ with $f \cdot \mathcal{C}_i = 0$ and gauge algebra $\mathfrak{g}_i \notin \{\varnothing, \mathfrak{su}_{2,3,4}, \mathfrak{sp}_2, \mathfrak{g}_2\}$ is an element of a little string theory (LST) for the charge class $f$.

3. The classification of SCFTs and LSTs, along with the unitarity of the *H*-string, places upper bounds on the number of tensor, vector and hypermultiplets.

---

Claim 1 corresponds to a simple geometrical fact that there always exists a rational

null curve $f$ for any smooth $B$ with Picard rank at least two. This follows from classical algebraic geometry facts (see e.g. [83]) that they have Kodaira dimension $-\infty$ and can always blow to a $\mathbb{P}^1$ fibration (except for $B = \mathbb{P}^2$), and we may pull back the fiber curve to $B$. This rational curve $f$ corresponds to the $H$-string, which is the key ingredient in our proof.

Claim 2 corresponds to the fact that any such $\mathcal{C}_i$ must map to a point along the composition $p : B \longrightarrow \mathbb{F}_n \longrightarrow \mathbb{P}^1$.[18] (In general, $\mathcal{C}$ covers $\mathbb{P}^1$ ($\mathcal{C} \cdot f$) times by a textbook result named projection formula [84].) These are the geometric counterparts of Sections 3.2 and 3.3.

Having seen that any elliptic Calabi-Yau with smooth $B \neq \mathbb{P}^2$ can be written as a two-step fibration $X \xrightarrow{\pi} B \xrightarrow{p} \mathbb{P}^1$, we wish to classify the singularity of these fibrations. This is the content of Claim 3. Classification of LSTs has two parts: classification of the base and classification of the fiber, which corresponds to singularities of $\pi$ and $p$ respectively. This classification is completely rigorous in algebraic geometry (although the frozen LSTs are not geometric). Note that the Picard rank $h^{1,1}$ of $B$ is given by the number of independent fibral $\mathbb{P}^1$ plus 1, the classification of singular fibers of $p$ place an upper bound on the Picard rank.

A key physical input, which is not even mentioned as part of the claim because it is obvious physically, is the gravitational anomaly cancellation condition $H - V + 29T = 273$. $V$ is the number of vector multiplets, with a non-Abelian part coming from the (codimension one) discriminant loci and the gauge group corresponding to their elliptic fiber[19], and an Abelian part coming from the Mordell-Weil group of the elliptic Calabi-Yau; $H$ is the number of hypermultiplets, with a charged part coming from codimension two singular fibers at the intersection of discriminant loci (also determined by gauge anomaly cancellation condition in physics) and an uncharged part coming from the complex structure moduli of the elliptic Calabi-Yau as well as its possible (terminal) singularities. Despite their intrinsic nature and great significance, *we do not have a geometric proof of the anomaly cancellation condition in full generality.* This is one of the key missing pieces to geometrize our physical proof of boundedness. Some special cases, for example, when we have only one non-Abelian gauge group, were proved in the series of papers [74, 85–87].[20] The basic idea is to calculate the Euler characteristic of Calabi Yau from the singular fibers. See also [88, 89] for an explanation of anomaly constraints in F-theory models by ensuring that no field-dependent Chern-Simons terms arise upon circle compactification. It is quite surprising that the existence of an $H$-string which is easy in geometry takes much more effort in physics, while anomaly cancellation which is automatic in physics seems more complicated in geometry.

An important observation in our proof is that only $E, F$-type gauge algebras (which come from $-n$ curves for $5 \leq n \leq 12$) contribute negatively to the left-hand side $H - V$, and we only need to bound the number of such curves $\mathcal{D}_i$ with exceptional gauge algebras. We treat two cases $f \cdot \mathcal{D}_i = 0$ and $f \cdot \mathcal{D}_i \neq 0$ separately. In the first case, $\mathcal{D}_i$ lies in a singular fiber of $p : B \longrightarrow \mathbb{P}^1$ and such singular fibers are classified in little string theory, although a

---

[18]We thank Sheldon Katz for providing this geometric proof of Claim 2.

[19]This correspondence is related to the fact that both elliptic fiber and Lie groups admit ADE classification.

[20]We thank Timo Weigand and Antonella Grassi for bringing these results to our attention.

single $\mathcal{D}_i$ might have a negative contribution to the gravitational anomaly, all $\mathcal{D}_i$ in the same singular fiber (or LST) have a positive contribution, and although there are families of LSTs with infinitely many elements, their contributions to gravitational anomaly diverge to infinity, hence only finitely many are embeddable in a compact manifold, and hence the number of $\mathcal{D}_i \cdot f = 0$ is bounded. In the second case, we need another fact that $\mathcal{D}_i$ produces current algebras in the standard $H$-string chiral algebra, which is another physics input from F-theory.

## 6  Discussion

We have argued, with minimal assumptions, that 6d supergravity theories with $\mathcal{N} = (1,0)$ supersymmetry admit a universal bound on the number of massless fields. Moreover, we have shown that the upper bounds we found for $T, V$ are saturated by specific points on the string landscape, reinforcing the belief in the string lamppost principle (SLP) that all UV-complete quantum gravity theories belong to the string landscape. This result, providing new mathematical predictions for the upper bounds on the Hodge numbers of elliptic Calabi-Yau 3-folds ($h^{1,1}, h^{2,1} \leq 491$ and $h^{1,1}(\text{Base}) \leq 194$), would be fascinating to derive purely from mathematical approaches, without relying on our physical arguments. Our results including the fact that the upper bounds on the number of massless fields are relatively small also reinforce the Swampland program: UV complete theories of quantum gravity are rather rare, and so, this implies strong restrictions on EFTs that admit a UV completion, which is the main aim of the Swampland program to distill.

There are a number of directions that would be natural to extend this work. While we have shown that the numbers of massless fields in 6d supergravities are finite, and consequently, the choices for matter representations are also finite for non-Abelian gauge factors, this does not imply that the matter representations for Abelian factors are similarly finite. There are infinitely many choices for $U(1)$-charges allowed by anomaly constraints [90] that would need to be trimmed to a finite set.

It would also be intriguing to extend these bounds to all theories with 8 supercharges in lower dimensions. This would mirror the development of such a bound for 16 supercharge theories, which are strongly constrained in the highest allowed dimension (d=10) due to anomalies, and were subsequently extended to lower dimensions [7, 91–96]. It has been found that the upper bounds for matter multiplets in lower dimensions arise from the 10d case upon toroidal compactification. This raises the natural question of whether this is the case for theories with 8 supercharges as well. A natural conjecture is that the upper bounds on vectors and hypers in lower dimensions are also obtained by circle compactifications of the maximal ones in 6d. Specifically, for $\mathcal{N} = 1$ in 5d, we expect $r(V) \leq 490, H \leq 492$; for $\mathcal{N} = 2$ in 4d, $r(V) \leq 491, H \leq 492$; for $\mathcal{N} = 4$ in 3d, $H, V \leq 492$. Indeed, the mirror symmetry for CY 3-folds [97] fits beautifully with Type II string realizations of these theories, and the convergence of these bounds in 3d fits elegantly with the expected mirror symmetry of $\mathcal{N} = 4$ theories in 3d [98].

Some evidence already supports this conjecture: The known upper bounds on the Hodge numbers of elliptic Calabi-Yau threefolds, which yield lower-dimensional theories

with eight supercharges through M-theory or type II compactifications, coincide with the elliptic ones, leading to the upper bounds above. Extending this program to achieve a full classification of supergravity theories with eight supercharges in lower dimensions would be highly interesting. A natural approach would be to apply the strategy used in this paper, combining global supergravity constraints from the emergence of local CFTs that define the boundaries of moduli space, an idea initiated in [62], could lead to the establishment of finiteness bounds for eight-supercharge supergravity theories.

The finiteness of the quantum gravity landscape, particularly the existence of a bound on the number of massless fields in quantum gravity theories, seems not to be too far out of reach: massless fields are not typically expected without supersymmetry. Moreover, the cases with less supersymmetry, such as 4d $\mathcal{N} = 1$ supergravity theories in the context of F-theory, are known to have a finite landscape due to the finiteness of elliptic CY 4-folds [78] and constraints on flux vacua [99–101]. In fact, even without relying on these, superpotentials are typically expected to eliminate massless fields [102], except for those descending from higher-supersymmetry theories, where boundedness can potentially be established.

Lastly, there is an intriguing aspect of BPS cone generators in 6d supergravity theories that merits further investigation. As proven in this paper, generators supporting gauge algebras $\mathfrak{g} \neq \mathfrak{g}_{\text{small}}$ that do not intersect an *H*-string are part of LSTs in the charge class for the *H*-string. In addition, we proved a stronger claim for geometric models: any generator, regardless of its associated gauge algebra, not intersecting a fiber must be part of LSTs in the fiber's charge class. We thus conjecture, though this is not used in this paper, that this stronger statement holds universally for all 6d supergravity theories, beyond the geometric setting. Proving this conjecture is left for future work.

## Acknowledgments

We would like to thank Yuta Hamada, Jonathan Heckman, Sheldon Katz, Gary Shiu, Houri Tarazi and Timo Weigand for useful discussions. H.K. is supported by Samsung Science and Technology Foundation under Project Number SSTF-BA2002-05 and by the National Research Foundation of Korea (NRF) grant funded by the Korean government (MSIT) (2023R1A2C1006542). The work of H.K. at Harvard University is supported in part by the Bershadsky Distinguished Visiting Fellowship. C.V. and K.X. are supported in part by a grant from the Simons Foundation (602883, CV) and the Della Pietra Foundation.

# A Structures of SCFTs and LSTs

The classification of SCFTs begins with the classification of the bases, which determines the intersection structure of tensor multiplets. In F-theory, this intersection structure translates into the geometric structure of a Kähler base $B$ in an elliptic fibration $X \to B$. The base is composed of tensor multiplets, some of which may support gauge algebras. Each elementary tensor multiplet in the base is assigned a self-intersection number $-n$ where $0 < n \leq 12$, and is referred to as *shrinkable*, meaning that, within the tensor branch, it is possible to reach a conformal fixed point where the BPS strings charged by the 2-form gauge field of the tensor multiplet become tensionless. The tensor multiplets must be arranged in such a way that they can all shrink simultaneously. This requires the intersection form $\Omega$ of the tensor multiplets to be negative-definite, which strongly constrains the structure of the base. Using this structure, the classification of 6d $\mathcal{N} = (1,0)$ superconformal field theories (SCFTs) and little string theories (LSTs) were thoroughly examined in [13–15, 103]. A concise review of the base structures and the classification of SCFTs and LSTs is presented in Section 3.4.

This appendix provides further details on the structures of SCFTs and LSTs used in our proof of the boundedness of 6d supergravity theories. To construct an SCFT, we begin with a base comprising a set of tensor multiplets arranged with a negative-definite intersection form $\Omega$. A general configuration for a base with more than five nodes is provided in (3.17). An LST base can then be formed by adding an extra tensor multiplet, introducing one null direction into the intersection form. Once the base is established, tensor multiplets can be decorated by assigning them gauge algebras and charged hypermultiplets in a way that cancels all gauge anomalies. For each tensor multiplet with a self-intersection $b_i^2 = -n$ where $0 \leq n \leq 12$, the allowed gauge algebra and charged hypermultiplets are listed in Table 4.

Next, each interior and side link forms either a linear chain or a T-shaped diagram, constructed by joining a finite number of non-DE type atoms. Notably, the interior link always takes the form of a linear chain, with $-1$ tensors at both ends. For instance, the following interior links $L_{i,i+1}$ connect two nodes $g_i$ and $g_{i+1}$, each with a gauge symmetry of types $D, E_6, E_7, E_8$.

$$
\begin{aligned}
&\mathfrak{so}_{2n} \overset{1,1}{\otimes} \mathfrak{so}_{2n} : \ \mathfrak{so}_{2n} 1 \mathfrak{so}_{2n} \ , &\quad &\mathfrak{e}_6 \overset{3,3}{\bigcirc} \mathfrak{e}_6 : \ \mathfrak{e}_6 1315131 \mathfrak{e}_6 \ , \\
&\mathfrak{e}_7 \overset{3,2}{\otimes} \mathfrak{so}_{2n} : \ \mathfrak{e}_7 1231 \mathfrak{so}_{2n} \ , &\quad &\mathfrak{e}_7 \overset{4,3}{\otimes} \mathfrak{e}_6 : \ \mathfrak{e}_7 12315131 \mathfrak{e}_6 \ , \\
&\mathfrak{e}_8 \overset{4,2}{\otimes} \mathfrak{so}_{2n} : \ E_8 12231 \mathfrak{so}_{2n} \ , &\quad &\mathfrak{e}_7 \overset{4,4}{\otimes} \mathfrak{e}_7 : \ \mathfrak{e}_7 123151321 \mathfrak{e}_7 \ , \\
&\mathfrak{e}_6 \overset{2,2}{\otimes} \mathfrak{e}_6 : \ \mathfrak{e}_6 131 \mathfrak{e}_6 \ , &\quad &\mathfrak{e}_8 \overset{5,3}{\otimes} \mathfrak{e}_6 : \ \mathfrak{e}_8 122315131 \mathfrak{e}_6 \ , \\
&\mathfrak{e}_7 \overset{3,3}{\otimes} \mathfrak{e}_7 : \ \mathfrak{e}_7 12321 \mathfrak{e}_7 \ , &\quad &\mathfrak{e}_8 \overset{5,4}{\otimes} \mathfrak{e}_7 : \ \mathfrak{e}_8 1223151321 \mathfrak{e}_7 \ , \\
&\mathfrak{e}_8 \overset{5,5}{\otimes} \mathfrak{e}_8 : \ \mathfrak{e}_8 12231513221 \mathfrak{e}_8 \ . &\quad & \\
\end{aligned}
\tag{A.1}
$$

Here, the superscripts $p, q$ above $\otimes$ and $\bigcirc$ denote the shifts in the intersection numbers for the adjacent $i$-th and $(i+1)$-th nodes, such that $(-n_i, -n_{i+1}) \to (-n_i + p, -n_{i+1} + q)$. The gauge algebras in the adjacent nodes can also take an E-type sub-algebra of the ones in

| $G_i$ | $H_i$ | $b_i^2$ | $b_0 \cdot b_i$ | Notes |
|---|---|---|---|---|
| $\mathfrak{g}$ | **Adj** | 0 | 0 | |
| $\mathfrak{su}_N$ | $(2N) \times \mathbf{N}$ | −2 | 0 | |
| $\mathfrak{su}_N$ | $(N-8) \times \mathbf{N} \oplus \frac{\mathbf{N(N+1)}}{\mathbf{2}}$ | −1 | −1 | $N \geq 8$ |
| $\mathfrak{su}_N$ | $(N+8) \times \mathbf{N} \oplus \frac{\mathbf{N(N-1)}}{\mathbf{2}}$ | −1 | 1 | |
| $\mathfrak{su}_N$ | $16 \times \mathbf{N} \oplus 2 \times \frac{\mathbf{N(N-1)}}{\mathbf{2}}$ | 0 | 2 | |
| $\mathfrak{su}_N$ | $\frac{\mathbf{N(N-1)}}{\mathbf{2}} \oplus \frac{\mathbf{N(N+1)}}{\mathbf{2}}$ | 0 | 0 | |
| $\mathfrak{su}_6$ | $15 \times \mathbf{6} \oplus \frac{1}{2}\mathbf{20}$ | −1 | 1 | |
| $\mathfrak{su}_6$ | $17 \times \mathbf{6} \oplus \mathbf{15} \oplus \frac{1}{2}\mathbf{20}$ | 0 | 2 | |
| $\mathfrak{su}_6$ | $18 \times \mathbf{6} \oplus \mathbf{20}$ | 0 | 2 | |
| $\mathfrak{su}_6$ | $\mathbf{6} \oplus \frac{1}{2}\mathbf{20} \oplus \mathbf{21}$ | 0 | 0 | |
| $\mathfrak{so}_N$ | $(N-8) \times \mathbf{N}$ | −4 | −2 | $N \geq 8$ |
| $\mathfrak{so}_N$ | $(N-7) \times \mathbf{N} \oplus (2^{\lfloor \frac{10-N}{2} \rfloor}) \times \mathbf{2}^{\lfloor \frac{\mathbf{N-1}}{\mathbf{2}} \rfloor}$ | −3 | −1 | $12 \geq N \geq 7$ |
| $\mathfrak{so}_N$ | $(N-6) \times \mathbf{N} \oplus (2 \times 2^{\lfloor \frac{10-N}{2} \rfloor}) \times \mathbf{2}^{\lfloor \frac{\mathbf{N-1}}{\mathbf{2}} \rfloor}$ | −2 | 0 | $13 \geq N \geq 6$ |
| $\mathfrak{so}_N$ | $(N-5) \times \mathbf{N} \oplus (3 \times 2^{\lfloor \frac{10-N}{2} \rfloor}) \times \mathbf{2}^{\lfloor \frac{\mathbf{N-1}}{\mathbf{2}} \rfloor}$ | −1 | 1 | $12 \geq N \geq 5$ |
| $\mathfrak{so}_N$ | $(N-4) \times \mathbf{N} \oplus (4 \times 2^{\lfloor \frac{10-N}{2} \rfloor}) \times \mathbf{2}^{\lfloor \frac{\mathbf{N-1}}{\mathbf{2}} \rfloor}$ | 0 | 2 | $14 \geq N \geq 4$ |
| $\mathfrak{sp}_N$ | $(2N+8) \times \mathbf{2N}$ | −1 | 1 | |
| $\mathfrak{sp}_N$ | $16 \times \mathbf{2N} \oplus (\mathbf{N-1})(\mathbf{2N+1})$ | 0 | 2 | |
| $\mathfrak{sp}_3$ | $\frac{35}{2}\mathbf{6} \oplus \frac{1}{2}\mathbf{14}'$ | 0 | 2 | |
| $\mathfrak{e}_8$ | | −12 | 10 | |
| $\mathfrak{e}_7$ | $\frac{k}{2} \times \mathbf{56}$ | $k-8$ | $k-6$ | $k \leq 8$ |
| $\mathfrak{e}_6$ | $k \times \mathbf{27}$ | $k-6$ | $k-4$ | $k \leq 6$ |
| $\mathfrak{f}_4$ | $k \times \mathbf{26}$ | $k-5$ | $k-3$ | $k \leq 5$ |
| $\mathfrak{g}_2$ | $(3k+1) \times \mathbf{7}$ | $k-3$ | $k-1$ | $k \leq 3$ |

**Table 4**. Gauge algebras $G_i$ and charged hypermultiplets $H_i$ supported on $b_i$ with $b_i^2 \leq 0$

this list. Additionally, if $p$ or $q$ is less than 4, the corresponding adjacent node can support a D-type gauge algebra, as long as the intersection pairing of the tensor multiplets remains negative definite. The superscript $p$ is referred to as minimal if $p = 1$ for $\mathfrak{so}$, $p = 2$ for $\mathfrak{e}_6$, $p = 3$ for $\mathfrak{e}_7$, and $p = 4, 5$ for $\mathfrak{e}_8$ gauge algebras on the left adjacent node. Similarly, $q$ is minimal if it satisfies the same condition for the right adjacent node. A link with both minimal $p$ and $q$ is called a *minimal link*, and otherwise, it is called a *non-minimal link*. The classification of all possible interior and side links, as well as the nodes $g_i$ and their gauge algebra decorations, is found in [13, 15, 17].

In constructing LSTs for $H$-strings that can be embedded within 6d supergravity, the gravitational anomaly contributions from the links and nodes play a critical role. Therefore, we calculate the gravitational anomaly contributions for all interior links, assuming minimal gauge algebras for both the interior links and adjacent nodes. For example, the interior link $\mathfrak{e}_8 \overset{5,5}{\otimes} \mathfrak{e}_8$, a key component in the construction of the theory hosting the maximum number of tensor multiplets, contains 11 tensor multiplets with $\mathfrak{f}_4 \times \mathfrak{g}_2^2 \times \mathfrak{sp}_1^2$ gauge algebras. The total gravitational anomaly contribution from this link, including half-contributions from

two adjacent nodes with $\mathfrak{e}_8$ gauge algebra, is

$$H + 29T - V = 16 + 29 \times (11) - (52 + 14 \times 2 + 3 \times 2) + (29 - 248) = 30 \,, \qquad \text{(A.2)}$$

where 16 is from the charged hypermultiplets of the $\mathfrak{g}_2$ and $\mathfrak{sp}_2$ gauge algebras, while 52, 14, and 3 correspond to the vector multiplet contributions from $\mathfrak{f}_4$, $\mathfrak{g}_2$, and $\mathfrak{sp}_1$, respectively. The last term, $29 - 248$, corresponds to the half-contribution from the two adjacent '$-12$' tensor nodes. Altogether, we find $\Delta(\mathfrak{e}_8 \overset{5,5}{\otimes} \mathfrak{e}_8) = 30$. Other results are summarized as follows:

$$\Delta(\mathfrak{e}_6 \overset{2,2}{\otimes} \mathfrak{e}_6) = 30\,, \ \ \Delta(\mathfrak{e}_6 \overset{2,2}{\otimes} \mathfrak{so}_8) = 55\,, \ \ \Delta(\mathfrak{e}_7 \overset{3,2}{\otimes} \mathfrak{so}_8) = 55.5\,, \ \ \Delta(\mathfrak{e}_6 \overset{3,2}{\otimes} \mathfrak{so}_8) = 83\,,$$

$$\Delta(\mathfrak{e}_7 \overset{3,3}{\otimes} \mathfrak{e}_7) = 30\,, \ \ \Delta(\mathfrak{e}_7 \overset{3,3}{\otimes} \mathfrak{e}_6) = 57.5\,, \ \ \Delta(\mathfrak{e}_7 \overset{3,3}{\otimes} \mathfrak{so}_8) = 82.5\,, \ \ \Delta(\mathfrak{e}_6 \overset{3,3}{\otimes} \mathfrak{e}_6) = 85\,,$$

$$\Delta(\mathfrak{e}_6 \overset{3,3}{\otimes} \mathfrak{so}_8) = 110\,, \ \ \Delta(\mathfrak{e}_8 \overset{4,2}{\otimes} \mathfrak{so}_8) = 27\,, \ \ \Delta(\mathfrak{e}_7 \overset{4,2}{\otimes} \mathfrak{so}_8) = 84.5\,, \ \ \Delta(\mathfrak{e}_6 \overset{4,2}{\otimes} \mathfrak{so}_8) = 112\,, \quad \text{(A.3)}$$

for the links that do not include a '$-5$' tensor multiplet, and

$$\Delta(\mathfrak{e}_6 \overset{3,3}{\otimes} \mathfrak{e}_6) = 86\,, \ \ \Delta(\mathfrak{e}_6 \overset{3,3}{\otimes} \mathfrak{so}_8) = 111\,, \ \ \Delta(\mathfrak{e}_7 \overset{4,3}{\otimes} \mathfrak{e}_7) = 115.5\,, \ \ \Delta(\mathfrak{e}_7 \overset{4,3}{\otimes} \mathfrak{so}_8) = 140.5\,,$$

$$\Delta(\mathfrak{e}_6 \overset{4,3}{\otimes} \mathfrak{e}_6) = 143\,, \ \ \Delta(\mathfrak{e}_6 \overset{4,3}{\otimes} \mathfrak{so}_8) = 168\,, \ \ \Delta(\mathfrak{e}_7 \overset{4,4}{\otimes} \mathfrak{e}_7) = 87\,, \ \ \Delta(\mathfrak{e}_7 \overset{4,4}{\otimes} \mathfrak{e}_6) = 114.5\,,$$

$$\Delta(\mathfrak{e}_6 \overset{4,4}{\otimes} \mathfrak{e}_6) = 142\,, \ \ \Delta(\mathfrak{e}_8 \overset{5,3}{\otimes} \mathfrak{e}_6) = 40\,, \ \ \Delta(\mathfrak{e}_7 \overset{5,3}{\otimes} \mathfrak{e}_6) = 97.5\,, \ \ \Delta(\mathfrak{e}_6 \overset{5,3}{\otimes} \mathfrak{e}_6) = 125\,,$$

$$\Delta(\mathfrak{e}_8 \overset{5,4}{\otimes} \mathfrak{e}_7) = 58.5\,, \ \ \Delta(\mathfrak{e}_8 \overset{5,4}{\otimes} \mathfrak{e}_6) = 86\,, \ \ \Delta(\mathfrak{e}_7 \overset{5,4}{\otimes} \mathfrak{e}_7) = 116\,, \ \ \Delta(\mathfrak{e}_7 \overset{5,4}{\otimes} \mathfrak{e}_6) = \Delta(\mathfrak{e}_6 \overset{5,4}{\otimes} \mathfrak{e}_7) = 143.5\,,$$

$$\Delta(\mathfrak{e}_6 \overset{5,4}{\otimes} \mathfrak{e}_7) = 87.5\,, \ \ \Delta(\mathfrak{e}_8 \overset{5,5}{\otimes} \mathfrak{e}_8) = 30\,, \ \ \Delta(\mathfrak{e}_8 \overset{5,5}{\otimes} \mathfrak{e}_7) = 87.5\,, \ \ \Delta(\mathfrak{e}_8 \overset{5,5}{\otimes} \mathfrak{e}_6) = 115\,,$$

$$\Delta(\mathfrak{e}_7 \overset{5,5}{\otimes} \mathfrak{e}_7) = 145\,, \ \ \Delta(\mathfrak{e}_7 \overset{5,5}{\otimes} \mathfrak{e}_6) = 172.5\,, \ \ \Delta(\mathfrak{e}_6 \overset{5,5}{\otimes} \mathfrak{e}_6) = 200 \,, \qquad \text{(A.4)}$$

for the links with a '$-5$' tensor multiplet. In this result, the $\mathfrak{e}_7$ and $\mathfrak{e}_8$ gauge algebra for adjacent nodes can be replaced by $\mathfrak{e}_7'$ and $\mathfrak{e}_8', \mathfrak{e}_8'', \mathfrak{e}_8'''$ algebras. However, such a replacement only increases the gravitational anomaly contributions, and can occur for the two leftmost and rightmost nodes when the base contains six tensors and above. Also, the minimal gauge algebras on specific tensor multiplets in these interior links and the $\mathfrak{so}_8$ gauge algebras on a node can be enhanced to non-minimal gauge algebras, but this too increases the anomaly contributions.

Another important factor for interior links (connected to adjacent nodes) is the sum of the number of tensor multiplets and the rank of gauge algebras, referred to as '$v$', which plays a critical role in establishing the bound on the rank of gauge algebras in a supergravity theory. For instance, the interior link $\mathfrak{e}_8 \overset{5,5}{\otimes} \mathfrak{e}_8$ with an adjacent node (half of two adjacent nodes) yields $v = 12 + (4 + 2 \times 2 + 1 \times 2) + 8 = 30$, where 12 counts the 12 tensor multiplets, the last 8 comes from the $\mathfrak{e}_8$ gauge algebra on the adjacent note, and and the middle term accounts for the rank of the gauge algebras in the link. Similarly, one can count these

numbers for all possible interior links as

$$v(\mathfrak{e}_6 \overset{2,2}{\otimes} \mathfrak{e}_6) = 12\,,\ \ v(\mathfrak{e}_6 \overset{2,2}{\otimes} \mathfrak{so}_8) = 11\,,\ \ v(\mathfrak{e}_7 \overset{3,2}{\otimes} \mathfrak{so}_8) = 13.5\,,\ \ v(\mathfrak{e}_6 \overset{3,2}{\otimes} \mathfrak{so}_8) = 13\,,$$

$$v(\mathfrak{e}_7 \overset{3,3}{\otimes} \mathfrak{e}_7) = 18\,,\ \ v(\mathfrak{e}_7 \overset{3,3}{\otimes} \mathfrak{e}_6) = 17.5\,,\ \ v(\mathfrak{e}_7 \overset{3,3}{\otimes} \mathfrak{so}_8) = 16.5\,,\ \ v(\mathfrak{e}_6 \overset{3,3}{\otimes} \mathfrak{e}_6) = 17\,,$$

$$v(\mathfrak{e}_6 \overset{3,3}{\otimes} \mathfrak{so}_8) = 16\,,\ \ v(\mathfrak{e}_8 \overset{4,2}{\otimes} \mathfrak{so}_8) = 15\,,\ \ v(\mathfrak{e}_7 \overset{4,2}{\otimes} \mathfrak{so}_8) = 14.5\,,\ \ v(\mathfrak{e}_6 \overset{4,2}{\otimes} \mathfrak{so}_8) = 14\,, \qquad \text{(A.5)}$$

for the links that do not include a '$-5$' tensor multiplet, and

$$v(\mathfrak{e}_6 \overset{3,3}{\otimes} \mathfrak{e}_6) = 22\,,\ \ v(\mathfrak{e}_6 \overset{3,3}{\otimes} \mathfrak{so}_8) = 21\,,\ \ v(\mathfrak{e}_7 \overset{4,3}{\otimes} \mathfrak{e}_7) = 25\,,\ \ v(\mathfrak{e}_7 \overset{4,3}{\otimes} \mathfrak{so}_8) = 23.5\,,$$

$$v(\mathfrak{e}_6 \overset{4,3}{\otimes} \mathfrak{e}_6) = 24\,,\ \ v(\mathfrak{e}_6 \overset{4,3}{\otimes} \mathfrak{so}_8) = 23\,,\ \ v(\mathfrak{e}_7 \overset{4,4}{\otimes} \mathfrak{e}_7) = 27\,,\ \ v(\mathfrak{e}_7 \overset{4,4}{\otimes} \mathfrak{e}_6) = 26.5\,,$$

$$v(\mathfrak{e}_6 \overset{4,4}{\otimes} \mathfrak{e}_6) = 26\,,\ \ v(\mathfrak{e}_8 \overset{5,3}{\otimes} \mathfrak{e}_6) = 26\,,\ \ v(\mathfrak{e}_7 \overset{5,3}{\otimes} \mathfrak{e}_6) = 25.5\,,\ \ v(\mathfrak{e}_6 \overset{5,3}{\otimes} \mathfrak{e}_6) = 25\,,$$

$$v(\mathfrak{e}_8 \overset{5,4}{\otimes} \mathfrak{e}_7) = 28.5\,,\ \ v(\mathfrak{e}_8 \overset{5,4}{\otimes} \mathfrak{e}_6) = 28\,,\ \ v(\mathfrak{e}_7 \overset{5,4}{\otimes} \mathfrak{e}_7) = 28\,,\ \ v(\mathfrak{e}_7 \overset{5,4}{\otimes} \mathfrak{e}_6) = v(\mathfrak{e}_6 \overset{5,4}{\otimes} \mathfrak{e}_7) = 27.5\,,$$

$$v(\mathfrak{e}_6 \overset{5,4}{\otimes} \mathfrak{e}_7) = 27\,,\ \ v(\mathfrak{e}_8 \overset{5,5}{\otimes} \mathfrak{e}_8) = 30\,,\ \ v(\mathfrak{e}_8 \overset{5,5}{\otimes} \mathfrak{e}_7) = 29.5\,,\ \ v(\mathfrak{e}_8 \overset{5,5}{\otimes} \mathfrak{e}_6) = 29\,,$$

$$v(\mathfrak{e}_7 \overset{5,5}{\otimes} \mathfrak{e}_7) = 29\,,\ \ v(\mathfrak{e}_7 \overset{5,5}{\otimes} \mathfrak{e}_6) = 28.5\,,\ \ v(\mathfrak{e}_6 \overset{5,5}{\otimes} \mathfrak{e}_6) = 28 \,, \qquad\qquad\qquad \text{(A.6)}$$

for the links with a '$-5$' tensor multiplet.

## B  Maximum rank configurations for semi-classical LSTs

Semi-classical LSTs are composed solely of '$-1$', '$-2$', and '$-4$' tensor multiplets, which support gauge algebras of *Sp*, *SU* and *SO*-type. When these LSTs are embedded into a supergravity theory, they accommodate high-rank gauge algebras with relatively fewer tensor multiplets, unlike LSTs containing '$-n$' tensor multiplets for $n > 4$. This makes it possible to find supergravity theories with the maximal rank of gauge algebras among those containing these specific LSTs.

The base configurations for semi-classical LSTs are classified as shown in (4.7) and (4.8). We will investigate all possible gauge algebras on these bases embedded in supergravity by utilizing a numerical optimization method to determine the maximal rank for each configuration. Our first step is to gauge the flavor symmetries in a semi-classical LST as much as possible by introducing external tensor and vector multiplets on them, while intentionally avoiding additional hypermultiplets. This step ensures minimal external contributions, $\Delta_{\text{ext}}$, to the gravitational anomaly, which allows the resulting theory to accommodate more tensor multiplets and higher-rank gauge algebras. We observe that higher-rank gauge algebras can be achieved when a flavor symmetry from a '$-1$' tensor multiplets is gauged by an external '$-4$' tensor multiplet with an *SO*-type gauge algebra, and a flavor symmetry from a '$-2$' tensor multiplets is gauged by an external '$-2$' tensor multiplet with an *SU*-type gauge algebra. Additionally, we find that when only a single LST is present in the charge class $f$ for the *H*-string, higher-rank gauge algebras can be achieved. By incorporating these external elements, we use numerical optimization to maximize $\mathcal{V}$,

the number of tensor multiplets plus the rank of gauge algebras, for supergravity theories that contain a semi-classical LST in the charge class $f$.

We implemented **Python** programs, gathered in a collection of companion files, to carry out the numerical optimization. In the following, we summarize the supergravity configurations that include each semi-classical LST with the highest rank of gauge algebras. Interestingly, we found that all semi-classical LSTs within the supergravity theories we identified realized by instantonic little strings in Heterotic string theories on ALE singularities, which were investigated in [29] and more recently in-depth in [60, 61].

Hereafter, we use $(n)$ to denote external '$-n$' tensor multiplets. Each $\mathfrak{su} - \mathfrak{su}$ and $\mathfrak{su} - \mathfrak{sp}$ pair includes a bi-fundamental hypermultiplet, while each $\mathfrak{so} - \mathfrak{sp}$ pair includes a bi-fundamental half-hypermultiplet. For the type **1**) base, the supergravity theory that yields the maximum rank of gauge algebras arises from

$$
\begin{array}{c}
\mathfrak{sp}_{40} \\
1 \\
\end{array}
$$

$$
\textbf{1)} \quad
\begin{array}{ccccccccc}
\mathfrak{so}_{64} & \mathfrak{sp}_{56} & \mathfrak{so}_{176} & \mathfrak{sp}_{72} & \mathfrak{so}_{128} & \mathfrak{sp}_{48} & \mathfrak{so}_{80} & \mathfrak{sp}_{24} & \mathfrak{so}_{32} \\
4 & 1 & 4 & 1 & 4 & 1 & 4 & 1 & (4)
\end{array} \ . \tag{B.1}
$$

This construction produces a supergravity theory featuring $T = 9$ tensor multiplets and 12 neutral hypermultiplets. The rank of the gauge algebras is $r(V) = 480$, giving a total $\mathcal{V}$ of 489. This theory describes the 24 point-like instantons in the $SO(32)/\mathbb{Z}_2$ Heterotic string theory sitting at an $E_8$ singularity on a K3 surface, as discussed in [29].

The maximal rank configuration for the supergravity with type **2**) base is

$$
\begin{array}{c}
\mathfrak{sp}_{28} \\
1 \\
\end{array}
$$

$$
\textbf{2)} \quad
\begin{array}{cccccccc}
\mathfrak{sp}_{12} & \mathfrak{so}_{64} & \mathfrak{sp}_{44} & \mathfrak{so}_{128} & \mathfrak{sp}_{48} & \mathfrak{so}_{80} & \mathfrak{sp}_{24} & \mathfrak{so}_{32} \\
1 & 4 & 1 & 4 & 1 & 4 & 1 & (4)
\end{array} \ . \tag{B.2}
$$

This theory has $T = 8$, a gauge algebra rank of $r(V) = 308$, and includes 13 neutral hypermultiplets. It is also realized by the 24 point-like instantons in the $SO(32)/\mathbb{Z}_2$ Heterotic string theory at an $E_7$ singularity, as studied in [29].

The configuration below represents the supergravity theory that reaches the maximal rank for the type **3**) base:

$$
\textbf{3)} \quad
\begin{array}{cccccc}
\mathfrak{su}_{32} & \mathfrak{su}_{64} & \mathfrak{sp}_{48} & \mathfrak{so}_{80} & \mathfrak{sp}_{24} & \mathfrak{so}_{32} \\
2 & 2 & 1 & 4 & 1 & (4)
\end{array} \ . \tag{B.3}
$$

This theory has $T = 5$, a gauge algebra rank of $r(V) = 222$ and 14 neutral hypermultiplets.

The maximal rank configuration for the supergravity with type **4**) base is

$$
\textbf{4)} \quad
\begin{array}{cccccc}
\mathfrak{su}_{16} & \mathfrak{su}_{32} & \mathfrak{su}_{48} & \mathfrak{su}_{64} & \mathfrak{sp}_{40} & \mathfrak{so}_{48} \\
(2) & 2 & 2 & 2 & 1 & 4
\end{array} \ . \tag{B.4}
$$

This theory has $T = 5$, a gauge algebra rank of $r(V) = 220$ and 12 neutral hypermultiplets.

The maximal rank configuration for the supergravity with type **5**) base is

$$
\textbf{5)} \quad
\begin{array}{cccc}
\mathfrak{su}_{16} & \mathfrak{su}_{32} & \mathfrak{sp}_{24} & \mathfrak{su}_{24} \\
(2) & 2 & 1 & 2
\end{array} \ , \tag{B.5}
$$

which has $T = 3$ with gauge algebras of rank $r(V) = 93$ and 15 neutral hypermultiplets.

The maximal rank configuration for the supergravity with type $\boldsymbol{6}$) base is

$$
\boldsymbol{6}) \quad \begin{array}{cccccccccccccc}
& & & \mathfrak{sp}_{16} & & & & & & & & & 1 & \\
& & & 1 & & & & & & & & & & \\
& \mathfrak{so}_{32} & \mathfrak{sp}_{24} & \mathfrak{so}_{80} & \mathfrak{sp}_{32} & \mathfrak{so}_{64} & \mathfrak{sp}_{24} & \mathfrak{so}_{48} & \mathfrak{sp}_{16} & \mathfrak{so}_{32} & \mathfrak{sp}_8 & \mathfrak{so}_{16} & \\
& (4) & 1 & 4 & 1 & 4 & 1 & 4 & 1 & 4 & 1 & 4 & 1
\end{array} \, .
\tag{B.6}
$$

This theory has $T = 13$, a gauge algebra rank of $r(V) = 256$ and 8 neutral hypermultiplets.

The maximal rank configuration for the supergravity with type $\boldsymbol{7}$) base is

$$
\boldsymbol{7}) \quad \begin{array}{cccccccccccc}
& & & \mathfrak{sp}_{16} & & & & & & & & \\
& & & 1 & & & & & & & & \\
& \mathfrak{so}_{32} & \mathfrak{sp}_{24} & \mathfrak{so}_{80} & \mathfrak{sp}_{32} & \mathfrak{so}_{64} & \mathfrak{sp}_{24} & \mathfrak{so}_{48} & \mathfrak{sp}_{16} & \mathfrak{so}_{32} & \mathfrak{sp}_8 & \mathfrak{su}_8 \\
& (4) & 1 & 4 & 1 & 4 & 1 & 4 & 1 & 4 & 1 & 2
\end{array} \, .
\tag{B.7}
$$

This theory has $T = 11$, a gauge algebra rank of $r(V) = 255$ and 9 neutral hypermultiplets.

The maximal rank configuration for the supergravity with type $\boldsymbol{8}$) base is

$$
\boldsymbol{8}) \quad \begin{array}{cccccccccc}
& & & \mathfrak{su}_{24} & & & & & & \\
& & & 2 & & & & & & \\
& \mathfrak{su}_{16} & \mathfrak{su}_{32} & \mathfrak{su}_{48} & \mathfrak{su}_{40} & \mathfrak{su}_{32} & \mathfrak{su}_{24} & \mathfrak{su}_{16} & \mathfrak{su}_8 & \\
& (2) & 2 & 2 & 2 & 2 & 2 & 2 & 2 & 1
\end{array} \, .
\tag{B.8}
$$

This theory has $T = 9$, a gauge algebra rank of $r(V) = 231$ and 3 neutral hypermultiplets.

Lastly, the maximal rank configuration for the supergravity with type $\boldsymbol{9}$) base is

$$
\boldsymbol{9}) \quad \begin{array}{cccccccc}
\mathfrak{so}_{32} & \mathfrak{sp}_{24} & \mathfrak{su}_{40} & \mathfrak{su}_{32} & \mathfrak{su}_{24} & \mathfrak{su}_{16} & \mathfrak{su}_8 & \\
(4) & 1 & 2 & 2 & 2 & 2 & 2 & 1
\end{array} \, .
\tag{B.9}
$$

This theory has $T = 7$, a gauge algebra rank of $r(V) = 155$ and 9 neutral hypermultiplets.

## C  Abelian gauge algebras

As discussed in the main content, we have already accounted for the potential presence of Abelian gauge algebras. Thus, the boundedness of the number of massless matter fields and the exact bounds we derived remain valid even when Abelian gauge algebras are included. The absence of Abelian gauge factors in cases with a large number of fields is simply due to the fact that Abelian gauge algebras cannot arise from any LSTs for $H$-strings, and each Abelian gauge factor provides only one vector multiplet which is the minimum possible number of vector multiplets while sacrificing the rank of gauge algebras by 1.

Abelian gauge factors can only appear on tensor multiplets associated with charges $b_i$ where $b_i^2 \geq 0$. Also, when $b_i^2 = 0$, indicating the absence of charged hypermultiplets, gauge anomaly cancellation conditions require $b_0 \cdot b_i = 0$. Hence, any LSTs associated to an $H$-string, where the charge $f$ has $f^2 = 0$ and $b_0 \cdot f = 2$, and any SCFTs cannot hold Abelian gauge algebras. This also means that any tensor charge $b_i$ that supports an Abelian gauge algebra must intersect with the charge class $f$ for the $H$-strings. This also includes those with $b_i^2 = 0$ and $b_0 \cdot b_i = 0$ as two distinct null charges must intersect.

As demonstrated, supergravity theories with $T \geq 1$ (except $T = 9$ and $b_0 = 0$ cases) contain an $H$-string in their BPS cones. Since the charges $b_i$ for Abelian gauge algebras

always intersect with the *H*-string, and considering the unitarity constraint from the worldsheet current algebra of the *H*-string, we conclude that the number of $U(1)$ gauge factors in 6d supergravity theories with $T \geq 1$ is bounded by 20 (and reduced to 4 in the cases where $T = 9$ and $b_0 = 0$). This bound coincides with the bound on Abelian sectors in F-theory models computed in [35], which was also generalized to supergravity theories assuming a specific lattice embedding of $\Omega$ and $b_0$, along with the existence of a null BPS string, though our result holds universally for any supergravity theories without relying on geometric considerations and particular lattice embeddings of anomaly vectors.

## D  BPS multiples of $b_0$

In Section 2.4 we argued that the charge class $Q = 12b_0$ always supports a BPS string. In F-theory this also follows from the Kodaira canonical bundle formula. Now, we construct F-theory examples to show that for any $m < 12$, $mb_0$ does not necessarily support a BPS string. It is worth noting that such examples are very rare. For instance, all toric bases and their blow-ups at boundary divisors satisfy the condition that $b_0$ is effective. In particular, every example that achieves the upper bounds we consider supports a BPS string in the class $b_0$. However, we will show that this property fails in the case of orbifolds derived from theories with higher supersymmetry.

We may first consider F-theory on the Z-orbifold $T^6/\mathbb{Z}_3$ [58], or its resolution by blowing up 27 singular points. It admits an elliptic fibration whose base is $T^4/\mathbb{Z}_3$ blow up 9 points.[21] Let's denote the exceptional divisor by $C_i$, then we have $b_0 = c_1 = \frac{1}{3}\sum C_i$. It is intuitively obvious that this cannot be written as a linear combination of integral divisors. We can give a rigorous proof for this: if $b_0$ is represented by a BPS cycle $C$, this would correspond to a globally defined holomorphic section $s$ of $\wedge^2 T_B$ on the base, which may be pulled back to $T^4$ along the $\mathbb{Z}_3$ cover. The pullback is not defined at the 9 fixed points, however, as they have codimension 2, they are removable singularities by Hartogs's extension theorem that holomorphic functions cannot have singularities of codimension higher than 1, and the holomorphic section can be extended over the whole $T^4$. So the pull-back of $s$ is also a globally defined section of $\wedge^2 T$ over $T^4$ and the only choice is a constant section. But the constant section is not invariant under the $\mathbb{Z}_3$ action, so we get a contradiction.[22] This argument generalizes to all examples studied in [104]. The $\mathbb{Z}_4 \times \mathbb{Z}_4$ example shows we need at least a factor of 4 and the $\mathbb{Z}_6 \times \mathbb{Z}_6$ example shows we need at least a factor of 6. So the factor of 12 for $b_0$ is indeed necessary to guarantee the existence of a BPS string.

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
