# Peer review of "Finite Landscape of 6d N=(1,0) Supergravity"

_SciPost Physics_

## Round 3 · Referee Report · Lara Anderson (Referee 1) · 2025-10-20

Strengths
- This paper is an important exploration of finiteness questions in the context of string compactifications and 3 (complex) dimensional Calabi-Yau manifolds. As a result, its results strengthen important synergistic links between geometry and string theory.
- In particular, past work (in both the mathematics and physics literature) has explored the question of finiteness of genus-one fibered Calabi-Yau manifolds. While past results established finiteness up to binational classes and certain bounds on the Hodge numbers of the Calabi-Yau threefold, this paper clarifies and solidifies these results and provides tighter bounds on the topological data.
- In general, the question addressed by this work is a central one in string compactifications and the string swampland program and this paper makes substantial new progress.
Weaknesses
- While not a weakness, the central finiteness arguments rely heavily on physical arguments derived from superconformal field theories and little string theories in 6-dimensions. It would be intriguing if some of these results could be independently derived in the context of differential/algebraic geometry in the future. However, this question is of course beyond the scope of the present work.
Report
Recommendation
Publish (surpasses expectations and criteria for this Journal; among top 10%)

---

## Round 3 · Referee Report · Washington Taylor (Referee 2) · 2025-11-24

Strengths
-
Overall, I think this is a very good paper and makes an important contribution to the literature, and I am happy to recommend it for publication by SciPost. The idea of the paper is to use some physically plausible assumptions to argue for finiteness of the set of 6D supergravity theories (and give specific bounds on the number of tensors and vectors). This is an important question; the question of finiteness of the string landscape has been studied in various contexts for many years, and 6D supergravity is an excellent place in which to nail down some concrete results on this.
-
On the whole I think the paper is well-written and the logic is good.
-
I believe the conclusions are correct.
-
I appreciate that the authors have spelled out clearly two of the main assumptions that they make, namely that the boundaries of tensor moduli space are given by BPS strings and LSTs, and that the set of such possibilities is already classified.
Weaknesses
1) There are some places in which I think the authors make additional assumptions, such as the structure of the BPS cone; see details below on requested changes
2) The article could be improved with some minor corrections; see details below on suggested changes
Report
Requested changes
1) My biggest concern about the paper is that in various places like section 2.1 and section 3, I believe that the authors implicitly make some additional assumptions, beyond those they state explicitly, that at least to my knowledge have not been clearly established anywhere in the literature. In general, the procedure that the authors follow in sections 2.1 and 3 is a physical translation of the geometric minimal model program. (I think it would actually be helpful if the authors make this correspondence a little more explicit, but that is perhaps a question of style). In proceeding with this analysis, however, it seems that they assume some things about the BPS cone that to me are not evident simply from the supergravity theory. In particular, it is not clear simply from the definition of the supergravity theory how one derives the BPS cone. While it is true that analyses such as those of their reference [41] describe explicitly certain BPS black string solutions, to my knowledge there is no systematic and complete description of the BPS string cone for a general 6D supergravity theory, in the absence of a geometric realization through something like F-theory. Thus, many of the features of this cone that the authors use, while plausible, and certainly evident in e.g. F-theory constructions, do not seem to be completely justified simply for a supergravity theory. For example, in equation (2.1), they define the tensor moduli space in terms of the cone of BPS strings. But, for example, it is not clear to me a priori that the set of BPS strings must always span a cone of dimension T +1, simply from the supergravity point of view. I think the paper would be strengthened if the authors can either justify their use of the BPS cone by actually defining this from the supergravity data in a well-defined way, or by including an additional explicit assumption about the structure of the BPS cone. Again, I suspect that all the features of the BPS cone that they use are correct, in part because these things all arise naturally when there is a geometric definition, but I don't see how this follows simply from supergravity.
2) The last sentence of the abstract says that this establishes the finiteness of the supergravity landscape for d >= 6. I think this is too strong, particularly given the second paragraph in the discussion section, where they point out at least one remaining issue, which is the possibility of infinitely many spectra for U(1)-charged matter. There is also the related issue of bounding discrete symmetry groups. These issues may be addressed at least in part by another very recent paper by the authors and other collaborators, but at least given what is in this paper I think that statement in the abstract is too strong.
3) I think there is a slightly cleaner argument for 2.4 that is less dependent upon physical assumptions: if a string is shrinkable there must be a place in the moduli space where j.C = 0, and since j.j > 0, it follows that anything orthogonal must have C.C < 0, since the total signature is (1, T). The authors are welcome to include this if they like.
4) There are a number of minor grammar issues in the draft that could be corrected, in particular missing articles. For example in the introduction beginning of paragraph 4, "... focus on the tensor branch...", middle of last paragraph of intro "particularly the H_string," 3rd paragraph of 2.1: "with non-zero tensor multiplets" perhaps should be "with a non-zero number of tensor multiplets". I suggest that the authors take a careful read through looking for such grammar issues.
5) I think there is something slightly off in the first sentence of the 2nd to last paragraph of the Discussion section: "The known upper bounds on the Hodge numbers of elliptic Calabi-Yau manifolds,... coincide with the elliptic ones". Probably the first elliptic should be "general"?
Recommendation
Publish (surpasses expectations and criteria for this Journal; among top 10%)

---

## Editorial Decision

unknown